# Targeting nucleic acid phase transitions as a mechanism of action for antimicrobial peptides

Tomas Sneideris[1,4], Nadia A. Erkamp [1,4], Hannes Ausserwöger [1,4], Kadi L. Saar [1], Timothy J. Welsh [1], Daoyuan Qian [1], Kai Katsuya-Gaviria [2], Margaret L. L. Y. Johncock [1], Georg Krainer [1], Alexander Borodavka [2] ✉ & Tuomas P. J. Knowles [1,3] ✉

Antimicrobial peptides (AMPs), which combat bacterial infections by disrupting the bacterial cell membrane or interacting with intracellular targets, are naturally produced by a number of different organisms, and are increasingly also explored as therapeutics. However, the mechanisms by which AMPs act on intracellular targets are not well understood. Using machine learning-based sequence analysis, we identified a significant number of AMPs that have a strong tendency to form liquid-like condensates in the presence of nucleic acids through phase separation. We demonstrate that this phase separation propensity is linked to the effectiveness of the AMPs in inhibiting transcription and translation in vitro, as well as their ability to compact nucleic acids and form clusters with bacterial nucleic acids in bacterial cells. These results suggest that the AMP-driven compaction of nucleic acids and modulation of their phase transitions constitute a previously unrecognised mechanism by which AMPs exert their antibacterial effects. The development of antimicrobials that target nucleic acid phase transitions may become an attractive route to finding effective and long-lasting antibiotics.

The overuse of antibiotics in both medicine and the food industries has led to the emergence of multidrug-resistant bacterial strains, which have become one of the major threats to human health, affecting millions worldwide. This situation has given rise to sustained efforts to find alternative solutions for combating bacterial infections. Antimicrobial peptides (AMPs) are a promising alternative to conventional small molecule drugs, as they have a broad range of antimicrobial activity and can target both Gram-positive and Gram-negative bacteria, as well as fungi[1–7].

AMPs are typically 10–25 amino acids long and carry a net positive charge. While it is generally thought that AMPs primarily target cell membranes via distinct mechanisms[8], many AMPs can cross the bacterial membrane without altering its integrity and interact with cytosolic targets such as proteins and nucleic acids. Most cell-penetrating AMPs are highly charged, with inclusions of hydrophobic residues[9]. The mechanism of action that underpins the antimicrobial activity of cell-penetrating AMPs is incompletely known, but specific patterns in sequence space that promote such activity are emerging. For example, arginine-rich AMPs are more prone to penetrate cell membranes compared with lysine-rich equivalents[9]. Moreover, peptides with a propensity to form alpha-helical structures upon binding to the membrane appear to be more cell penetrating compared to the ones displaying disordered conformations[9]. Similarly, cyclic AMPs are commonly more potent membrane penetrators compared to linear peptides[9]. Crucially, however, little is known about the intracellular mechanisms of action of AMPs. Several AMPs have

[1]Yusuf Hamied Department of Chemistry, University of Cambridge, Lensfield Road, Cambridge, UK. [2]Department of Biochemistry, University of Cambridge, Tennis Court Road, Cambridge, UK. [3]Cavendish Laboratory, Department of Physics, University of Cambridge, J J Thomson Ave, Cambridge, UK. [4]These authors contributed equally: Tomas Sneideris, Nadia A. Erkamp, Hannes Ausserwöger. ✉e-mail: ab2677@cam.ac.uk; tpjk2@cam.ac.uk

been shown to lead to intracellular granulation of bacterial cells[10] or induce nucleoid DNA condensation[11,12]. Both of these observations point towards changes in the localisation and phase states of nucleic acid molecules in bacteria upon interaction with AMPs. Another intriguing example is Buforin-2, which has been demonstrated to bind DNA and RNA in vitro[13–16] and was shown to possess lysis-free bactericidal activity, suggesting that the interactions between the peptide and nucleic acids may be a key to its mechanism of action. In the context of eukaryotic cells, proline-arginine-rich peptides and LL-III peptide were shown to affect the phase separation of phase separating proteins like Fused in Sarcoma (FUS) or P-granule protein LAF-1, enhancing their phase separation propensity or the ageing of preformed condensates[17,18].

Phase separation, a process well-established in polymer chemistry yet until recently rarely observed in vivo, is now broadly recognised as an important phenomenon governing the formation of membraneless organelles (MLOs), including the nucleolus, BR-bodies and stress granules[19–22]. Phase separation is emerging as a key principle controlling subcellular organisation, shown to be involved in a variety of biological processes including RNA metabolism, ribosome biogenesis, DNA damage response and signal transduction in cells across kingdoms of life[23–35]. During phase separation, biopolymers in solution demix and form a condensed liquid phase that possesses material properties distinct from those of the surrounding dilute phase[23,27]. While phase separation has been explored relatively widely across eukaryotes, only a few studies have focused on examining condensates in prokaryotic organisms due to their smaller sizes, which makes the probing of subcellular structures challenging[23]. However, MLOs formed through phase separation have been proposed to play a major role in the subcellular organisation of bacteria[23,30,36–38]. Indeed, membraneless compartments containing enzymes of many biochemical pathways have recently been found to be present in bacteria[23,30,36]. Notably, bacterial ribonucleoprotein bodies (BR-bodies) bring together the RNA degradosome machinery and its RNA targets, thereby providing bacteria with the possibility to locally control mRNA decay by forming condensates[37,38]. This suggests that modulation of the phase transition of bacterial nucleic acids may be a vital regulatory mechanism of bacterial growth.

Here, we investigate the mechanism of action of AMPs with respect to their ability to modulate phase transitions of nucleic acids. We applied a recently developed machine learning algorithm, trained to predict phase separation of proteins and peptide-nucleic acid complexes[39], and identified a set of AMPs exhibiting a high propensity to undergo phase separation in the presence of nucleic acids. For three representative AMPs, Buforin-2, P113 and Os-C, we demonstrate their ability to modulate phase transitions of nucleic acids resulting in the formation of biomolecular condensates. Unexpectedly, quantitative analyses of the phase behaviour of these AMPs reveal a dependence between the potency of AMPs to undergo phase separation with nucleic acids and their ability to inhibit prokaryotic transcription and translation in vitro. Finally, we show that the addition of these peptides to bacterial cells leads to the formation of intracellular foci-like condensates containing nucleic acids and AMPs. We propose that AMP-driven phase transitions of bacterial nucleic acids may contribute to the bacteriocidal activity of these peptides, in addition to their membrane destabilisation activities.

## Results and discussion
### Phase separation propensity of AMPs
We retrieved the sequences of previously identified AMPs deposited into the Database of Antimicrobial Activity and Structure of Peptides (DBAASP)[40] and in the APD3 antimicrobial peptide database[41] to analyse their predicted propensities to undergo phase separation. Interestingly, we find that some AMPs possess features[42–44] that are common amongst biopolymers prone to undergo phase separation[27,45].

The features include (i) the presence of charged amino acid residue patches; (ii) enrichment in polar amino acids (average polar amino acid fraction in AMPs is 20–30%); (iii) the presence of hydrophobic amino acid residues; (iv) the ability to oligomerise or/and aggregate; (v) and the lack of three-dimensional structure or the presence of intrinsically disordered regions.

To quantify the propensities of individual AMPs to undergo homotypic or nucleic acid-mediated phase separation, we analysed their amino acid sequences using the recently described machine learning approach DeePhase[39]. This approach classifies polypeptides using a combination of natural language processing approaches and physical features to train a machine learning model on datasets of proteins and peptides with different propensities to undergo phase separation. We extended this approach by retraining the model to account for protein interactions with nucleic acids (see 'Methods'), and used the models to examine both homotypic and heterotypic nucleic acid-mediated phase separation propensity on both the full human proteome (Fig. 1a), which contains the majority of well-characterised proteins known to phase separate, and a set of known AMPs (Fig. 1b).

The machine learning analysis reveals that a substantial fraction of AMPs are predicted to phase separate either alone (i.e. homotypic phase separation) or in the presence of nucleic acids (Fig. 1b). For instance, ~27% ($n$ = 3480) of AMPs have homotypic phase separation propensity scores above 0.5, suggesting that these peptides are predicted to be prone to undergoing phase separation. Notably, the phase separation propensity is displayed on a 0-1 scale, where 0 and 1 correspond to a low and a high phase separation propensity, respectively. Interestingly, for nucleic acid-mediated phase separation, almost 62% ($n$ = 8092) of AMPs have an LLPS propensity score above 0.5, while ~7% ($n$ = 944) of AMPs are predicted to have scores that exceed those of the highest predicted scores within the human proteome. Together, these results suggest a high probability of phase separation for the analysed AMPs. The larger fraction of AMPs (67% vs 27%) predicted to undergo phase separation in the presence of nucleic acids is in line with the observations that nucleic acids often facilitate phase transitions of nucleic acid-binding proteins[27].

Interestingly, compared to the human proteome (Fig. 1a), AMPs display distinct homotypic and nucleic acid-mediated phase separation propensity profiles. In the case of homotypic phase separation, the human proteome phase separation propensity is relatively evenly distributed over the whole propensity score range with two main peaks located at very low and very high propensity score values (Supplementary Fig. 1a). In contrast, for AMPs the distribution of homotypic phase separation propensity is significantly narrower with several peaks mainly centred around intermediate score values (Supplementary Fig. 1a). This suggests that AMPs have a lower propensity to undergo homotypic phase separation compared to well-characterised proteins that form condensates, such as FUS[46], TDP-43[47], $\alpha$-synuclein ($\alpha$SYN)[48] or ubiquilin-2 (UBQLN2)[49]. The tendency of AMPs to undergo homotypic LLPS, however, is still relatively high when compared to the human proteome.

Furthermore, in the presence of nucleic acids, a substantial fraction of AMPs have nucleic acids-mediated phase separation propensity values higher than that of the human proteome (Fig. 1b, and Supplementary Fig. 1b), including well-characterised proteins FUS, TDP-43, G3BP1[50] and HMGA1[51] known to phase separate with DNA/RNA. This suggests that many AMPs may have a high propensity to undergo phase separation in the presence of nucleic acids. To eliminate any bias due to length differences between AMPs (21 ± 13 amino acid residues, AA) and the human proteome (556 ± 548 AA) on the phase separation propensity profiles, we additionally screened two sets of unique random AA sequences of variable lengths (Supplementary Fig. 1). The results show that the chain length alone is unlikely to account for such distinct phase separation propensity profiles, further suggesting that the observed differences may be attributed to the unique AA

composition of AMPs. Interestingly, the majority of AMPs with phase separation propensity scores higher than 0.5 are 10–25 AA residues in length with a net charge of +5 to 10 and are slightly hydrophilic with polar amino acid fractions ranging from just a few to 20% (Supplementary Fig. 2). Notably, we do not observe a simple correlation between the net charge alone and the nucleic acids-mediated phase separation score, suggesting some sequence specificity and potentially highlighting the importance of charge-patterning in this process (Supplementary Fig. 2).

### AMPs form biomolecular condensates with nucleic acids in vitro

Given the predicted high propensity of AMPs to phase separate with nucleic acids from our machine learning analysis, we then probed their behaviour in vitro. We selected three representative AMPs, namely P113, Os-C and Buforin-2 (Table 1, Fig. 1b) all of which have been reported to interact with DNA/RNA[10–16,52] and have distinct predicted phase separation propensities (Fig. 1b).

To explore whether the three AMPs undergo phase separation and form liquid-like assemblies in vitro, we used fluorescently labelled peptides and probed their phase behaviour in the presence of a simple polyadenylic acid homopolymer (poly(A) RNA), lacking secondary structure. Using this system, we first examined the chemical parameter space and the material properties of the condensed phase. At a physiological salt concentration (150 mM KCl, 1 mM MgCl$_2$, see 'Methods') mixtures of three AMPs and poly(A) RNAs at 1:1 and 1:5 peptide:poly(A) mass ratios spontaneously demixed forming spherical liquid-like droplets (Fig. 2a). The AMP:poly(A) droplets exhibited a high degree of roundness >0.9 (Supplementary Fig. 3), suggesting that they are liquid-

like. Indeed, multiple incidents of droplets fusing and relaxing into larger ones were observed (Fig. 2b, Supplementary Movie 1, 2 and 3), with a characteristic relaxation time of ≈3 min. To investigate if the AMPs diffuse within the droplets, we analysed fluorescence recovery after photobleaching (FRAP) of FITC-AMPs in droplets (Fig. 2c and Supplementary Fig. 4). The half-life of fluorescence recovery, $\tau_{1/2}$, for a 1 μm$^2$ area in P113-poly(A) RNA condensates was 14 ± 2 s ($n = 4$ biological replicates). In the case of Os-C, and Buforin-2-poly(A) condensates, the $\tau_{1/2}$ values were 5 ± 1 s ($n = 4$ biological replicates) and 13 ± 2 s ($n = 4$ biological replicates), respectively. The shorter $\tau_{1/2}$ indicates higher mobility of fluorescent molecules (i.e. labelled AMPs) within the condensate and confirms that the condensates are liquid-like.

We then investigated the nature of the interactions between the AMPs and nucleic acid molecules within the condensates. To understand the relative contributions of electrostatic vs. hydrophobic interactions that drive phase separation, we examined the behaviour of the condensates in different chemical environments. First, we introduced 5% w/v of 1,6-hexanediol, an aliphatic alcohol disrupting hydrophobic interactions[53], into a sample with preformed condensates. No evident effect on AMP/total yeast RNA condensates was observed (Fig. 2d and Supplementary Fig. 5), suggesting that hydrophobic interactions do not play a significant role in condensate formation. We next examined the contribution of electrostatics by elevating the KCl concentration in the sample. To rule out the possibility of homotypic phase separation of homopolymeric poly(A) RNA in the presence of high KCl concentrations[54,55], we used total yeast RNA instead. Remarkably, at high KCl concentrations (500 mM) preformed

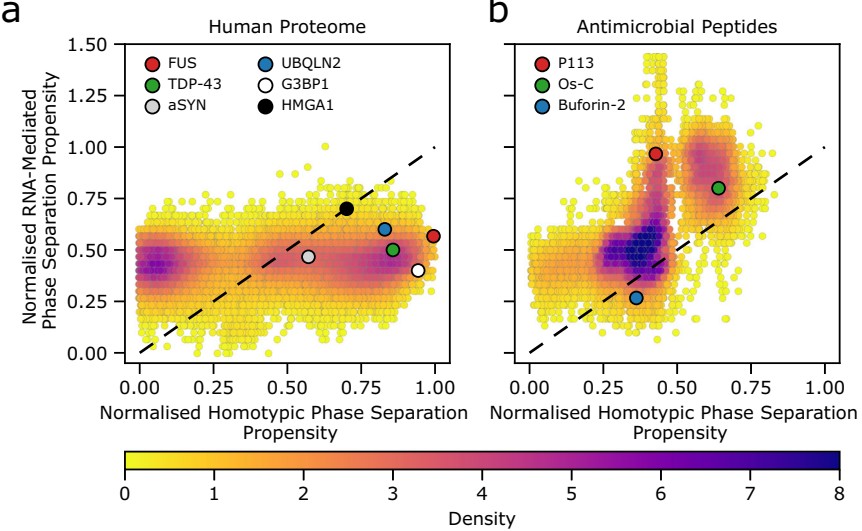

**Fig. 1 | The predicted phase behaviour of the human proteome and AMPs.** Homotypic phase separation propensity (x-axis) and RNA-mediated phase separation propensity (y-axis) for human proteome (**a**, 20324 proteins; UniProt) and a set of 13170 AMPs (**b**)[40]. The colour bar represents the overall phase separation propensity score density. Several proteins (e.g. FUS) known to undergo LLPS are shown as reference points. The diagonal black lines separate regions of

amino acid chains with a higher propensity to undergo phase separation in the presence of nucleic acids compared to homotypic phase separation propensity (above the line), while regions below the line represent nucleic acid-mediated phase separation being less favourable than homotypic phase separation. Source data are provided as a Source data file.

**Table 1 | Amino acid sequences and physicochemical properties of AMPs under investigation**

| Name | Sequence | Net charge | Gravy index | Polar Frac. | Cationic Frac. | Anionic Frac. | Aromatic Frac. |
|---|---|---|---|---|---|---|---|
| Os-C | KGIRGYKGGYKGAFKQTKY | +5.6 | −1.316 | 0.368 | 0.316 | 0.000 | 0.211 |
| P113 | AKRHHGYKRKFH-NH$_2$ | +4.8 | −2.283 | 0.083 | 0.667 | 0.000 | 0.167 |
| Buforin-2 | TRSSRAGLQFPVGRVHRLLRK | +5.3 | −0.638 | 0.333 | 0.333 | 0.000 | 0.048 |

The net charge and GRAVY (grand average of hydropathicity) index of AMPs as well as the polar, cationic, anionic and aromatic fractions of AA within the AMPs were obtained from sequence analysis via DeePhase.

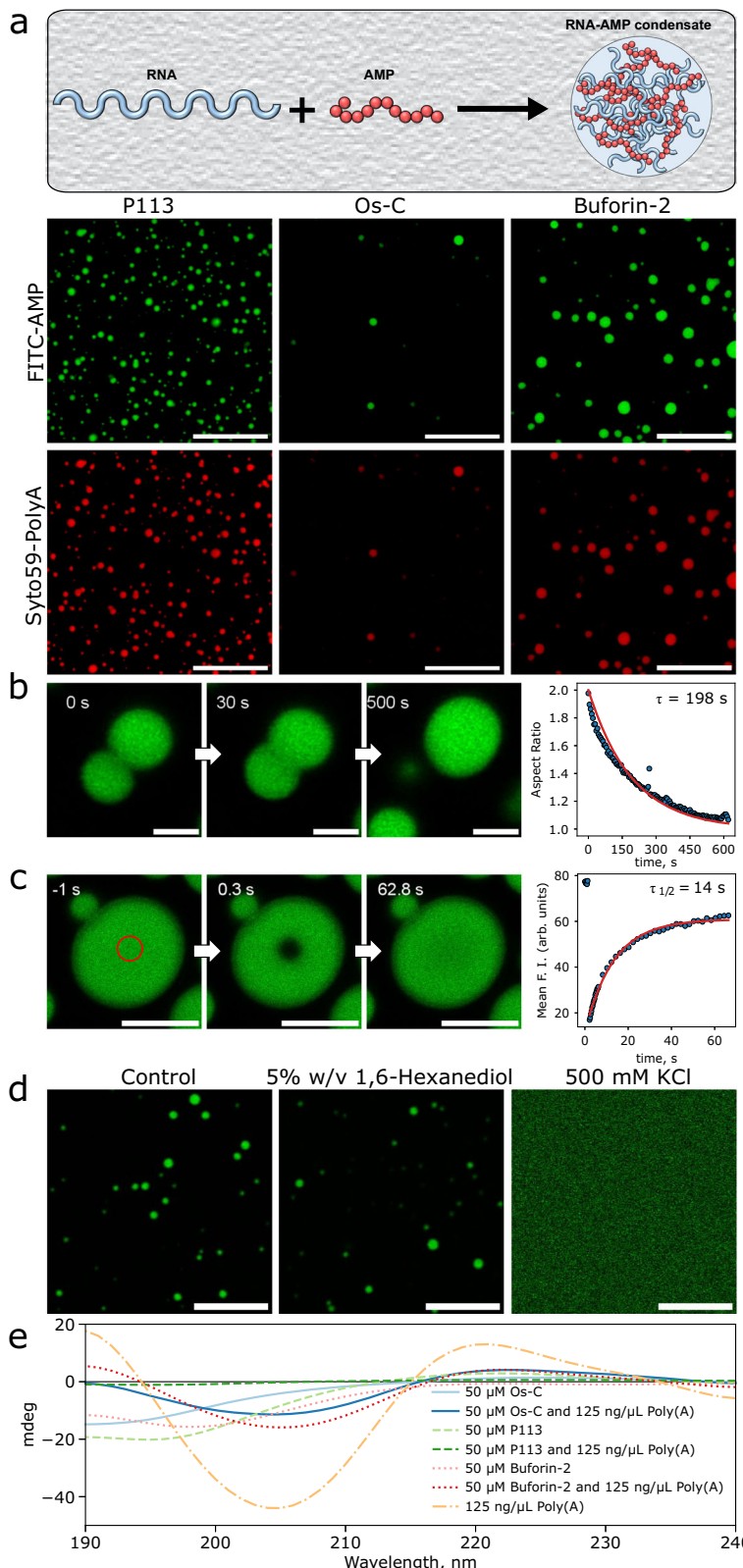

condensates were fully dissolved (Fig. 2d and Supplementary Fig. 5), underscoring the key role of electrostatics driving AMP-nucleic acid coacervation.

In aqueous solution, the majority of AMPs are disordered, however, upon interaction with biological membranes, they can acquire structure, including forming α-helices[56]. We were intrigued to see if AMPs display distinct structural features inside biomolecular condensates. Both, AMPs under investigation and poly(A) RNA alone, displayed a disordered structure in 10 mM sodium phosphate buffer solution pH 7.3 supplemented with 150 mM Na⁺ (Fig. 2e). When both components were present, condensates formed (Supplementary Fig. 6), however, we did not observe any secondary structure signatures. The findings are in line with the hypothesis that, at least in freshly formed condensates, disordered proteins or low-complexity

**Fig. 2 | Nucleic acid-binding AMPs P113, Os-C and Buforin-2 form liquid-like condensates with RNA. a** A schematic illustration depicting AMP-RNA condensate formation and the representative confocal microscopy images of P113-, Os-C and Buforin-2-poly(A) RNA condensates formed in 5% w/v PEG (20 kDa), 150 mM KCl, 1 mM MgCl$_2$, 50 mM HEPES pH 7.3. As described in 'Methods', FITC-labelled P113, Os-C, and Buforin-2 are shown in green and Syto59-labelled poly(A) RNA in red. The condensates shown were formed using the following concentrations of components: 300 μM P113 and 2000 ng/μL poly(A) RNA; 300 μM Os-C and 2000 ng/μL poly(A) RNA; 400 μM Buforin-2 and 1000 ng/μL poly(A) RNA. **b** Confocal

microscopy images of P113-poly(A) RNA condensates fusing. The plot on the right shows the dynamics of two droplets fusing and relaxing into a single droplet, with a characteristic relaxation time $\tau$ of ≈198 s. **c** FRAP analysis of P113-poly(A) RNA condensates with the characteristic FRAP recovery rate $\tau_{1/2}$ of 14 ± 2 s ($n$ = 4 biological replicates). **d** Confocal microscopy images of P113-total yeast RNA (see 'Methods') condensates before and after the introduction of 1,6-hexanediol (5%w/v) or the addition of 500 mM KCl. **e** Circular dichroism (CD) spectra of AMPs in mixed and demixed states. The scale bars are 10, 2, 5 and 10 μm for panels (**a**), (**b**), (**c**) and (**d**), respectively. Source data are provided as a Source data file.

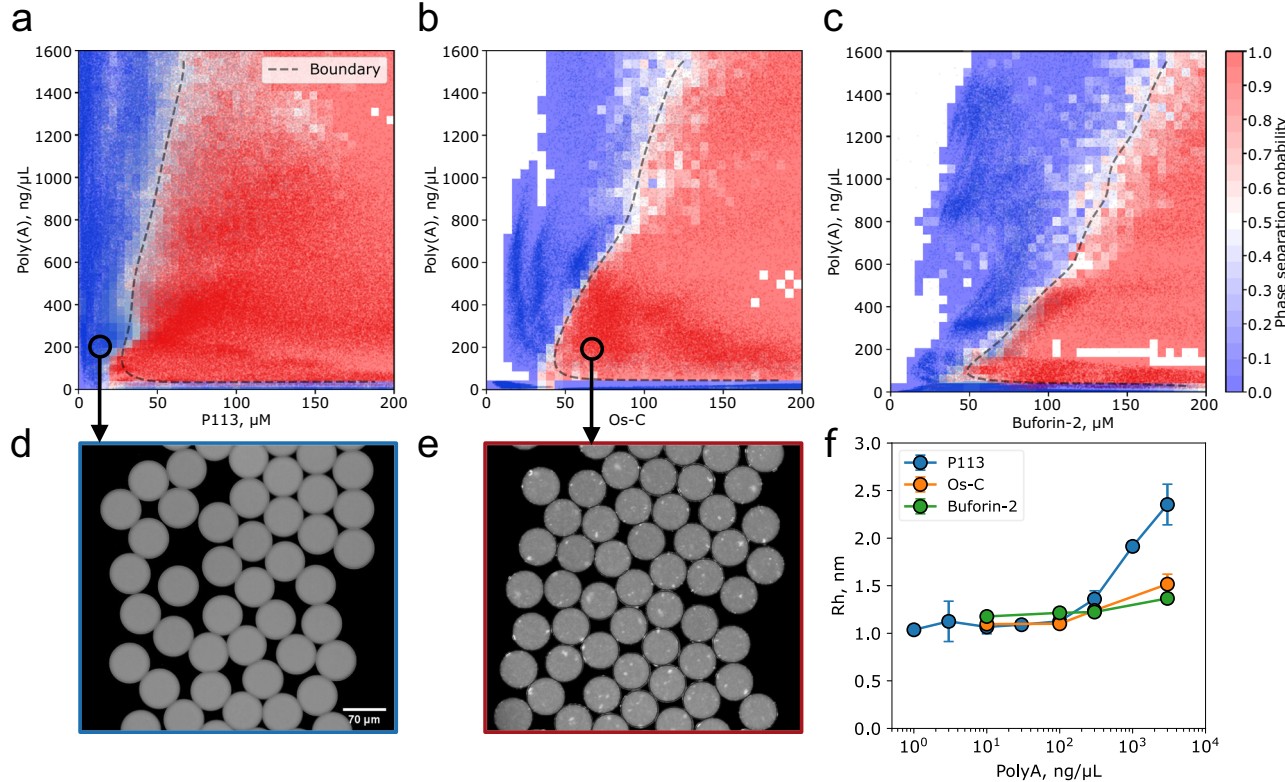

**Fig. 3 | Phase separation behaviour of AMPs.** Phase diagram for P113 (**a**), Os-S (**b**) and Buforin-2 (**c**) as a function of poly(A) RNA concentration. Blue points and red points correspond to the mixture being homogenous or phase separated, respectively. The colour-coded heat map shows the estimated phase separation probability over the range of AMP vs. poly(A) concentrations. The boundary (dashed line) highlights the region where the phase separation probability is equal to 0.5 and serves as a guide for the eye. The number of individual microenvironments

(individual data points) investigated: $n_{P113}$ = 246955 (**a**); $n_{Os-C}$ = 84398 (**b**); $n_{Buforin-2}$ = 167030 (**c**). Representative images of microfluidic droplets with individual microenvironments where no phase separation events were observed (**d**) and where phase separation events were observed (**e**). The hydrodynamic radius of FITC-AMPs as a function of poly(A) concentration measured at 1 μM AMP concentration (**f**). Data are presented as mean values ± SD ($n$ = 3 technical replicates). Source data are provided as a Source data file.

domains within the phase-separating proteins remain entirely disordered and dynamic inside the condensates[57].

## Quantification of AMP phase separation propensity

We next sought to elucidate the phase behaviour of these peptides through the measurement of phase diagrams at varying concentrations of AMPs and nucleic acids. Such phase diagrams can be time-consuming to generate at high resolution using conventional laboratory approaches due to the large number of concentrations of the components that have to be probed. Therefore, we took advantage of our combinatorial microdroplet platform PhaseScan[58] to generate high-density phase diagrams constructed from > 60,000 individual measurements.

Using PhaseScan, we first determined the conditions under which AMP/RNA mixtures undergo a transition from a homogeneous solution to a two-phase system (Fig. 3). No homotypic phase separation events were observed when either the RNA or the peptide were present

alone, suggesting that both poly(A) RNA and AMPs have to be present for phase separation to take place for all three AMPs tested. This result is consistent with the machine learning analysis described above which predicts a higher phase separation propensity for the heteromolecular system relative to the peptide alone. The location of the phase boundary suggests that relatively low concentrations of poly(A) RNA and specific AMPs are required for detectable condensate formation (Fig. 3) when both components are present in solution together. For a constant concentration of peptide, upon increasing the concentration of RNA, we first observe a transition from the homogeneous solution to the two-phase region; further increase in the RNA concentration leads to a reentrant transition back to a homogeneous solution phase. Such a reentrant transition[59] is often observed for multicomponent systems[60–62] including RNA-binding proteins[63]. Specifically, by drawing a vertical line at 50 μM for P113 (Fig. 3a), it becomes clear that at low poly(A) RNA concentrations ([RNA] ≤ 10 ng/μL), the solution is homogeneous, whereas increasing poly(A) RNA concentration results

in spontaneous demixing of the system into two phases: biopolymer-poor and a polymer-rich phase (Fig. 3e) (10 ng/µL ≤ [RNA] ≤ 1200 ng/µL). A further increase in poly(A) RNA ([RNA] > 1200 ng/µL) results in the dissolution of condensates via the reentrant phase transition.

Interestingly, the phase separation behaviour of all three AMPs was different. The phase diagrams and specifically the slopes of the high RNA reentrant boundary revealed that the P113 peptide was the most potent peptide-modulator of poly(A) RNA phase separation, followed by Os-C and Buforin-2. Notably, these experimental results were in good agreement with the theoretical predictions from our machine learning model (Fig. 1b), showing that P113 has the highest propensity to undergo phase separation with nucleic acids, while Buforin-2 has the lowest propensity amongst all three tested. Interestingly, the lower RNA concentration at which phase separation occurs is relatively constant for all three AMPs tested, indicating that at subsaturated RNA to AMP stoichiometries all three peptides form condensates. Yet, the distinct amino acid composition of the AMPs tested imparts specificity that dictates the overall stoichiometry and defines the amount of RNA that a condensate can accommodate without being dissolved in a reentrant transition driven by the binary interactions between the peptide and the RNA molecule.

Since all three AMPs displayed slightly different efficiencies in modulating poly(A) phase separation, we next sought to elucidate whether this effect could originate from the different affinities of the peptides for poly(A). To investigate this aspect further, we characterised the binary interaction between fluorescently labelled AMPs and poly(A) using microfluidic diffusional sizing (MDS)[64] (Fig. 3f). Specifically, we determined the hydrodynamic radius ($r_H$) of AMPs at varying poly(A) concentrations where no phase separation events were observed. Binding events are then observed due to changes in $r_H$ based on complexation between AMP and poly(A) RNA[64]. Based on the $r_H$ change, the onset of the binary interaction with poly(A) RNA follows the order P113 > Os-C > Buforin-2 (Fig. 3f). The trend matches perfectly the phase separation behaviour of peptides.

**Condensate-forming AMPs compact structured RNA**

Having examined the AMP-poly(A) interactions and condensate formation, we next investigated how P113, Os-C and Buforin-2 interact with structured, biologically relevant RNAs. We monitored AMP-induced phase separation of 16S and 23S rRNAs (Fig. 4a, b and Supplementary Figs. 7–13). Similarly to poly(A) RNA, P113, Os-C and Buforin-2 induced the condensation of 16S and 23S rRNAs (Fig. 4a, and Supplementary Figs. 7–13). Notably, depending on the AMP:RNA ratio, we also saw irregularly shaped condensates (Supplementary Figs. 10–12) that resembled those formed in the presence of poly(proline-arginine) peptide repeats[65], suggesting that the formation of a less fluid condensates might be a feature of more structured nucleic acids.

Next, we mapped the phase behaviour of the P113-23S rRNA system using the PhaseScan approach (Fig. 4b). Similarly to poly(A) RNA, we observed that relatively low concentrations of 23S rRNA and P113 were required to induce phase separation. As was found to be the case with poly(A) RNA, the slope of the phase boundary was steep with P113-23S rRNA, indicating that P113 is a potent modulator of RNA-induced phase separation.

To shed light on the mechanism of RNA-AMP condensation, we investigated AMP binding to 16S rRNA using MDS (Fig. 4c). Under physiological salt conditions, fluorescently labelled 16S rRNA has the $r_H$ of ~17 nm, consistent with previously determined values[66]. In the presence of AMPs (<50 µM), we observed an initial collapse of $r_H$ of 16S rRNA to ~5 nm. This suggests that AMPs can induce compaction of individual RNA chains similarly to other polyamines, e.g., spermidine (Sp$^{3+}$)[66]. Due to the small molecular weight of AMPs (~2 kDa), we were unable to detect the initial AMP binding events to the large RNA (~530 kD) that would not significantly contribute to the change in $r_H$ of

AMP-bound RNA. Remarkably, despite a similar net charge, the degree of RNA compaction by AMPs was much higher compared to Sp$^{3+}$ (i.e., 5.5 ± 0.6 nm for Buforin-2 versus 11.6 ± 0.25 nm for Sp$^{3+}$, at 50 µM of each polycation) supporting the notion that RNA compaction by AMPs cannot be simply explained by charge neutralisation, and suggesting specificity of binding. Similarly, the $r_H$ of longer RNAs including 23S rRNA, mRNA encoding the beta subunit of RNA polymerase (RpoB) and MS2 phage genomic ssRNAs decreased in the presence of different concentrations of Os-C (Supplementary Fig. 14). The change in $r_H$ of these RNAs with increasing Os-C concentration follows the same trend as in the case of 16S rRNA. Together, these results suggest that AMPs do not have a marked preference for individual transcripts, although some AMPs appear to be more potent at inducing RNA compaction followed by their condensation.

Hence, we hypothesized that nucleic acid-binding AMPs could over-compact nucleic acids and cause their condensation at µM concentrations. Indeed, above 50 µM of AMPs, the $r_H$ of the RNAs started to increase with increasing AMP concentration. Interestingly, such behaviour was not observed for Sp$^{3+}$, even at very high (>1 mM) concentrations. Based on the change of $r_H$ upon increasing AMP concentration, P113 shows the strongest effect on RNA compaction, followed by Os-C and Buforin-2. Similarly, the increase in $r_H$ for 16S rRNA was also more prominent initially in the presence of P113 followed by Os-C and Buforin-2. Microscopy images of these samples (Fig. 4a) revealed the formation of 1–8 µm spherical droplets at the AMP concentrations at which we observed an increase in the $r_H$ of 16S rRNA. In contrast, only relatively small RNA clusters (<1 µm in diameter), similar to those seen at low AMP concentrations before condensate formation, were observed even when Sp$^{3+}$ reached 500 µM. Taken together, these data reveal that nucleic acid-binding AMPs modulate RNA phase transitions, inducing transcript compaction at low concentrations prior to phase separation with RNA.

To further characterise AMP-RNA interactions in the condensed phase, we turned to measurements of the tie-lines of this system using a recently developed approach[67], which relates the partitioning of molecular species between dilute and dense phases directly to information on the emergent collective interactions. To achieve this objective, we measured the P113 concentration in the dilute phase varying 23S rRNA concentrations using photon-counting fluorescence microscopy (Fig. 4d), as described in 'Methods'. This analysis revealed that at low RNA concentrations, the dilute phase peptide concentration remains roughly constant, while it drops drastically at higher RNA concentrations, indicating peptide redistribution from the mixed phase to the dense phase. More specifically, at 125 µM P113 and 100 nM 23S rRNA, the dilute phase concentration of the peptide drops to roughly 55 µM, suggesting that a large fraction of AMPs remain in the dilute phase, and thus AMPs could still bind available RNA species outside the condensed phase. The tie-line gradient between both components can be extracted after the onset of phase separation, revealing the molar ratio of 23S rRNA to P113 of ~2.5 × 10$^{-4}$ (Fig. 4d). The observed positive gradient also suggests that both species will actively partition into condensates and display attractive interactions within the condensed phase[67], with the condensed phase maintaining a large molar excess of the peptide compared to RNA (~4000 peptides per RNA). Together, these results further support the model in which a large number of the AMPs act as transient stickers that cross-link the RNA strands. Considering the molecular weight of both species, the mass ratio within condensates is much more comparable (~5:1 for the peptide to RNA by mass) suggesting that both species are critical for generating the condensate scaffold. Thus, the tie-line analysis shows that AMP/RNA phase separation results in the sequestration of RNAs within the formed condensates. Taken together, our data suggest that AMPs can engage RNAs by compacting them followed by phase separation into dilute and condensed phases (Fig. 4e).

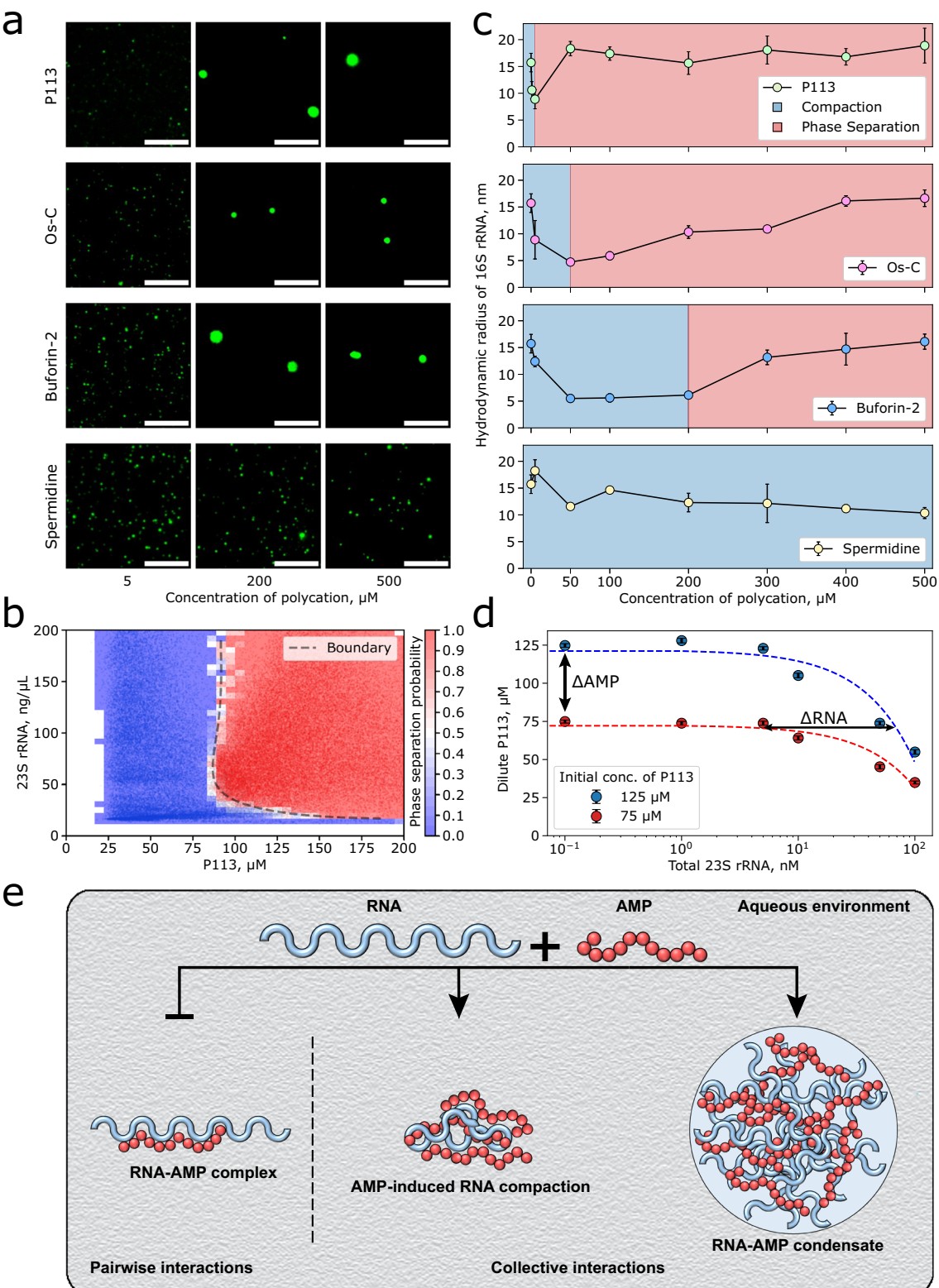

**Fig. 4 | AMPs compact and modulate phase separation of structured RNAs.**
**a** Widefield fluorescence microscopy images of 0.2 nM of 647N-labelled 16S rRNA in the presence of several concentrations of different polycations. The scale bars are 10 μm. **b** Phase diagram for P113 peptide as a function of 23S rRNA concentration (*n* = 60739 individual droplets measured). **c** Measurements of $r_H$ of 16S rRNA as a function of P113, Os-C, Buforin-2 or Sp³⁺ concentration. Data are presented as mean values ± SD (*n* = 3 technical replicates). MDS measurements were performed using 0.2 nM Atto 647N-labelled 16S rRNA. **d** Measurements of P113 concentration in dilute phase at varying 23S rRNA concentrations. Data are

presented as mean values ± SD (*n* = 3 biological replicates). Dashed lines are linear fits. The gradient of tie-line, $k = \frac{\Delta RNA}{\Delta AMP}$, provides information on whether solutes prefer to be in the same (positive gradient) or different (negative gradient) phases. In the current case, the gradient was positive, $-2.5 \times 10^{-4}$ 23S rRNA/P113 molar ratio, indicating that both species actively partition into condensates and exhibit attractive interactions within the condensate. **e** Schematic illustration depicting the formation of distinct AMP-RNA assemblies that are held together via interactions of different strengths and nature. Source data are provided as a Source data file.

## The mechanism of action of DNA/RNA-targeting AMPs

We hypothesized that the observed AMP-modulated phase separation of RNAs would result in the sequestration of the genetic material affecting bacterial viability. To understand better how AMP-induced nucleic acid phase separation would interfere with biological processes, we carried out cell-free protein synthesis (CFPS) reactions in the presence and absence of AMPs. For CFPS, we used a plasmid DNA containing a T7 promoter (Fig. 5a–c) or mRNA (Fig. 5d–f) that encodes an enhanced green fluorescent protein (eGFP). We monitored translation alone (mRNA) or both transcription and translation (using plasmid DNA) by measuring the production of eGFP. In the presence of all AMPs, the amount of eGFP decreases with increasing AMP concentration. Remarkably, P113 displayed the strongest and Buforin-2 the weakest effect on the translation. Fitting of the plasmid DNA- and mRNA-programmed CFPS reactions kinetics data to a simple model describing transcription and translation (see 'Methods') revealed that the observed inhibition of translation by P113 was more prominent when the reaction was programmed with mRNA (Fig. 5g and Supplementary Fig. 15), whereas Os-C and Buforin-2 displayed a slightly stronger effect on the template DNA-initiated CFPS reaction. Nevertheless, P113 had the greatest effect on eGFP production in both the mRNA and plasmid DNA-initiated CFPS reactions (Fig. 5g). To test whether the AMPs also inhibited transcription reaction, we also performed an in vitro transcription (IVT) reaction in the presence of various concentrations of AMPs (Supplementary Fig. 16). Again, P113 showed a strong inhibitory effect on the transcription reaction, while Buforin-2 had no significant effect on the efficacy of transcription under the same conditions (Supplementary Fig. 16).

It is likely that all AMPs under investigation interact with both DNA and mRNA and interfere with transcription and translation. The amplitude of the inhibitory effects of increased AMP concentrations in the CFPS reaction solution was similar to that when the amount of template DNA or mRNA was limited (Supplementary Fig. 16a, b). In both cases, the amount of protein produced decreased exponentially with decreasing concentrations of template DNA or mRNA (Supplementary Fig. 16d and e, respectively); however, the reaction rate did not change (Supplementary Fig. 17a, b). Similarly, in the presence of AMPs, the CFPS reaction rates are mostly unaffected with only P113 displaying a slight effect at high concentrations (Supplementary Fig. 17c–h). To rule out the possibility of eGFP quenching by the AMPs, we also added AMPs to the CFPS reaction once the fluorescence intensity plateaued (Supplementary Fig. 16c), further confirming that the observed effects are due to the inhibition of translation. Moreover, if AMPs targeted ribosomes or RNA polymerase directly, we would expect the reaction rate to drop drastically. However, since AMPs did not display a strong effect on the CFPS reaction rates, it is likely that AMPs primarily sequester nucleic acids through modulation of their phase separation.

To learn more about how AMPs compete against RNA/DNA-binding proteins we performed experiments where we first formed AMP-poly(A) RNA condensates and then added bacterial poly(A)-binding protein Hfq[68] (Case #1) or pre-incubated Hfq with poly(A) to enable formation of protein-RNA complex and then introduced AMPs into the sample (Case #2) (Fig. 5h and Supplementary Figs. 18–19). In the first case, at the P113:Hfq molar ratios ≥1, the P113-poly(A) condensates remained intact. Increasing Hfq concentration further resulted in a gradual decrease of P113-poly(A) condensates. Similarly, Os-C-poly(A) RNA and Buforin-2-poly(A) RNA condensates remained intact at the AMP:Hfq molar ratios ≥3 and started dissolving at higher Hfq concentrations. In the second case, P113-poly(A) RNA condensates could still form at P113:Hfq molar ratios ≥10, and Os-C-poly(A) RNA and Buforin-2-poly(A) RNA condensates formed at AMP:Hfq molar ratios ≥30. Thus, assuming the dissociation constant of the Hfq-poly(A) RNA complex ~ 30 nM[69], after AMP-RNA condensates are formed, high concentrations of tightly RNA-binding proteins are required to

outcompete AMPs and to disrupt the condensates. In the opposite case, at certain AMP:RNA-binding protein molar ratios, AMPs are also able to displace protein-bound RNA and induce its phase separation.

## Nucleic acid-binding AMPs are present in condensate-like clusters inside bacteria

Several AMPs have been reported to cause granulation inside bacteria[10–12]. We hypothesised, therefore, that such AMP-induced granulation could be explained by the phase separation of nucleic acids and AMPs. To test this hypothesis, we incubated E. coli cells with FITC-labelled AMPs, as described in 'Methods', followed by their imaging (Fig. 6a and Supplementary Figs. 20–27). As expected, all tested AMPs interacted with bacterial cells, with fluorescence signal increasing inside cells, suggesting that these peptides were able to translocate across the membrane, in line with previous reports[10,12,14,16]. We also noted that some cells contained AMP-positive foci-like clusters (Fig. 6a). Remarkably, these foci also stained positive for nucleic acids, suggesting that these clusters may represent biomolecular condensates comprised of AMPs and nucleic acids. We also noted that these structures were not observed in all cells. This variation is not unexpected because multiple factors can influence how AMPs translocate into cells and this process, in turn, would affect the intracellular AMP concentration. Furthermore, we expect the metabolic and transcriptional rates of individual cells to vary, resulting in different amounts of nucleic acids available to participate in AMP condensate formation. Indeed, the overall FITC-AMP signal intensity was higher in cells containing foci-like clusters compared to the ones lacking AMP-positive clusters. Similarly, the concentration of specific types of RNA molecules present (e.g. rRNA, mRNA, ncRNA, tRNA etc.) can affect the size of individual condensates formed, most of which would remain challenging to image via conventional microscopy given the sub-micrometre sizes of the majority of condensates observed in bacterial cells. Moreover, fewer condensate fusion events would also contribute to significantly smaller condensate sizes in bacterial cells, thus likely being an additional factor altering the apparent abundance of detectable condensates in AMP-treated cells. Condensates formed through LLPS can be metastable, hence, they can transition into glassy or gel-like states that do not necessarily display classical liquid properties[45]. The latter states, however, are reached through LLPS. This mechanism could in fact contribute towards cell death as it may provide cytotoxic pathways impossible for the cell to counteract. Components arrested in a gel-like state are typically unable to fuse forming large condensates. Thus, if the dynamic arrest happens shortly after the phase transition, the assemblies are generally too small to resolve by conventional confocal microscopy[45]. We cannot exclude the possibility that the observed AMP inclusion contain additional components beyond nucleic acids. It is possible that these clusters represent aggresomes[70] formed in response to AMP-induced cell stress, or naturally occurring bacterial condensates like BR-bodies or RNAP clusters[23]. Regardless of their nature, FITC-labelled AMPs colocalised with the observed clusters suggesting local high concentrations of AMPs within. Thus, despite the observed scarcity of such inclusions in bacterial cells, our data suggest that the formation of such clusters is concomitant with the addition of labelled AMPs.

To better understand the nature of sub-micron-sized foci-like structures that we observed inside bacterial cells, we investigated AMP-induced condensates formed in cell extract-based transcription/translation system derived from E. coli (Fig. 6b and Supplementary Figs. 28–29). Probing sub-micron-sized structures in bacterial cells is generally challenging, mainly due to their relatively small size[23]. The use of a cell extract-based system enabled us to overcome some limitations providing the opportunity to investigate the nature of AMP-induced condensates at near physiological conditions. Having an open system also meant that we could easily introduce phase separation modulators such as 1,6-hexanediol or KCl and investigate their effect on condensates. To learn more about the physical state of the

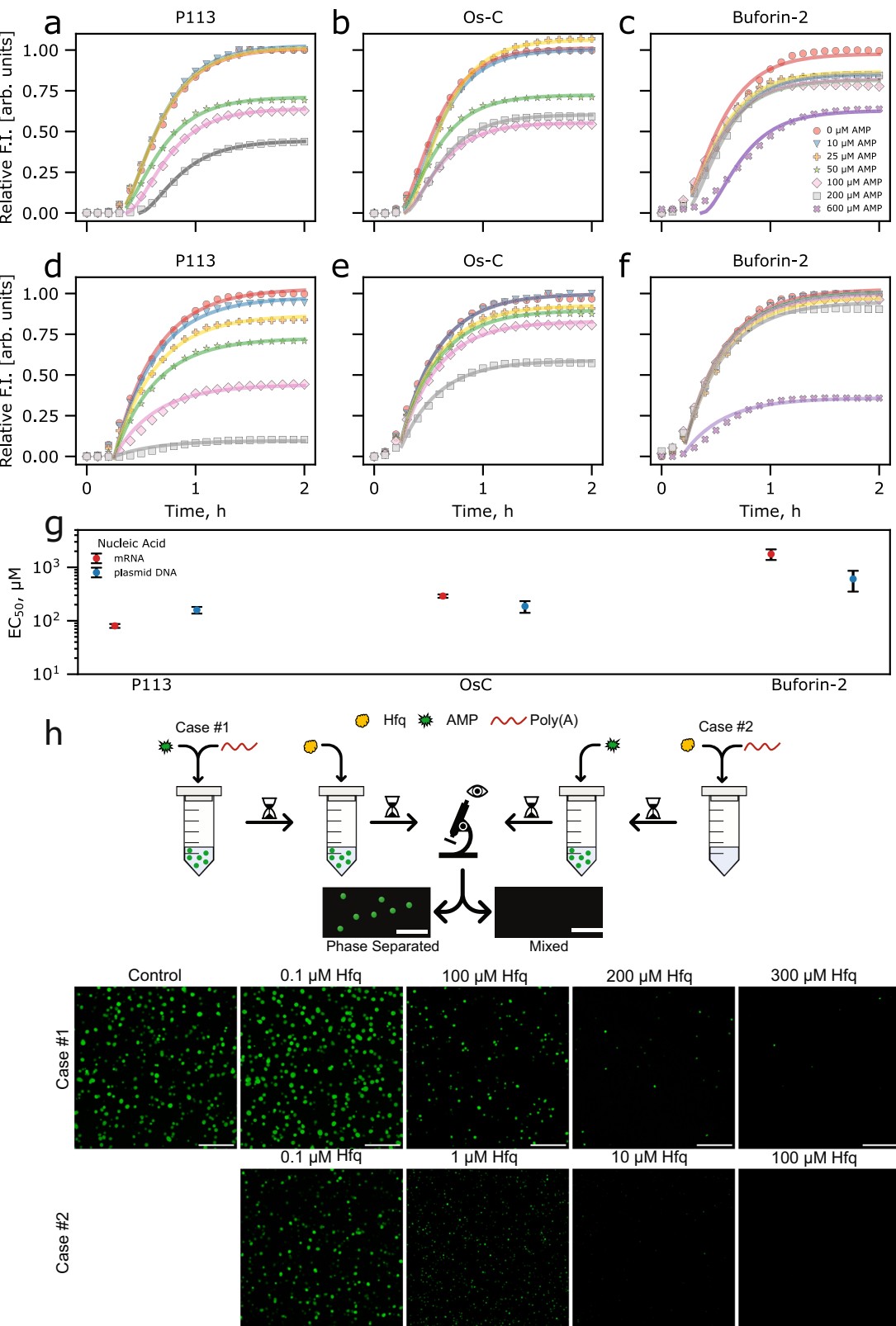

**Fig. 5 | AMP effects on transcription and translation processes.** Cell-free eGFP synthesis followed in the absence and presence of antimicrobial peptides (**a**–**f**). The cell-free synthesis of eGFP was initiated by supplementing CFPS solution with 6 ng/μL plasmid DNA (**a**–**c**) or 80 ng/μL mRNA (**d**–**f**) encoding eGFP protein. Continuous lines are the fits (see 'Methods'). Comparison of EC$_{50}$ values of AMPs in plasmid DNA- or mRNA-initiated CFPS reaction (**g**). Data are presented as the best-fit values ± RMSD. AMP vs. Hfq competition assay (**h**). Case #1: 100 μM P113 was mixed with 250 ng/μL of poly(A) and incubated for 5 min at room temperature. Subsequently, 0.1–300 μM Hfq was introduced and the sample was incubated for an additional 5 min before imaging via a confocal fluorescence microscope. Case #2: 0.1–100 μM Hfq was pre-incubated with 250 ng/μL of poly(A) for 5 min before introducing 100 μM P113. The scale bars are 20 μm. Source data are provided as a Source data file.

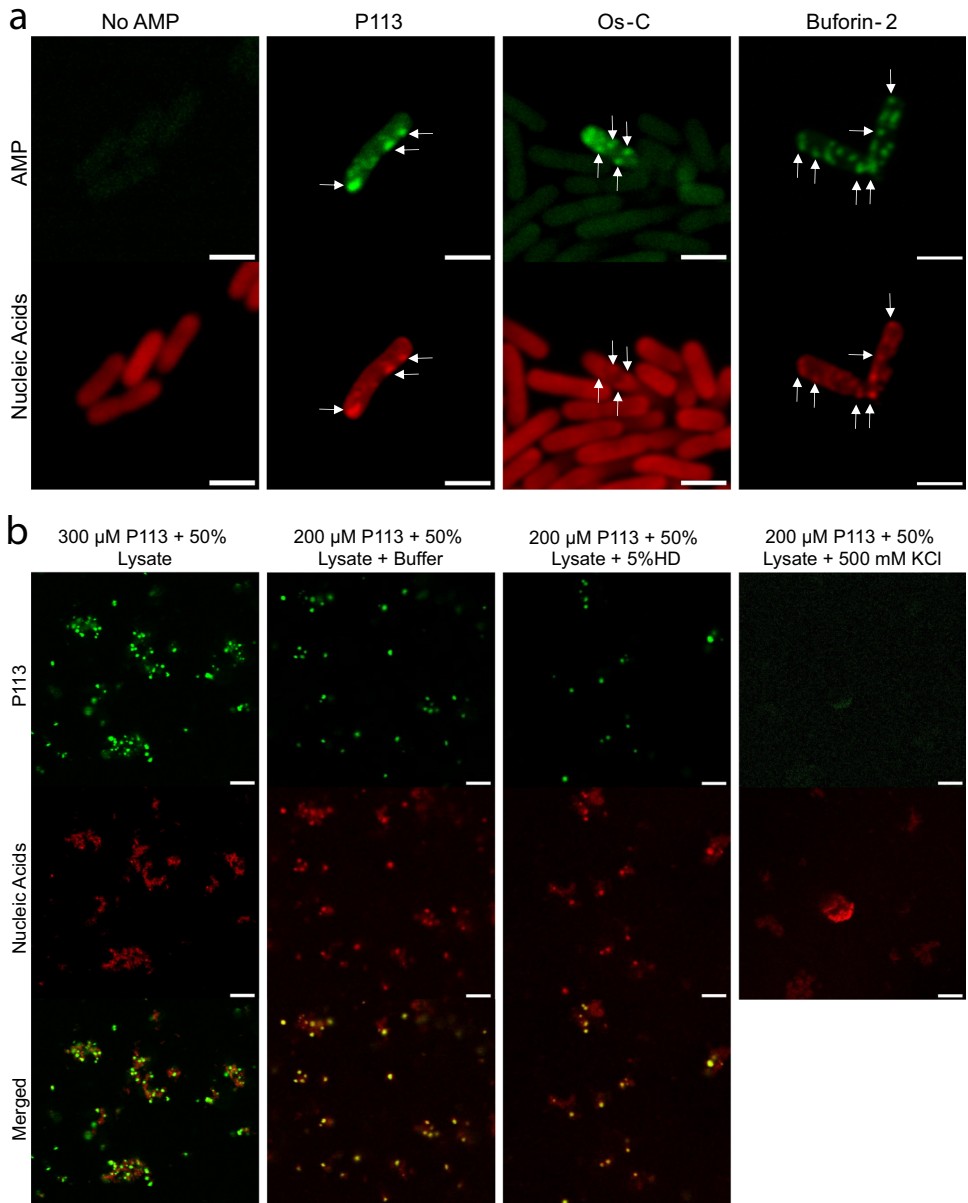

**Fig. 6 | AMPs induce foci-like condensate formation in bacterial cells and an extract-based active transcription/translation system derived from *E. coli.*** **a** Confocal microscopy images of *E. coli* cells incubated with 20 μM of FITC-P113, FITC-Os-C or FITC-Buforin-2 (green). Nucleic acids were stained with Bacto View Red (red). White arrows serve as a guide for the eye to help to indicate foci-like condensates. The control sample shows weak autofluorescence as biological samples tend to show it when excited at ~488 nm. The scale bars are 2 μm. **b** Confocal microscopy images of P113-induced condensates formed in a cell extract-based system. Nucleic acids were stained with Syto59 (red). The scale bars are 5 μm.

condensates we performed FRAP experiments on condensates formed in the lysate (Supplementary Fig. 30). The half-life of fluorescence recovery, $\tau_{1/2}$, for a ~1 μm$^2$ area in P113 condensates was $18 \pm 6$ s ($n = 5$ biological replicates). In the case of Os-C, and Buforin condensates, the $\tau_{1/2}$ values were $1 \pm 0.4$ s ($n = 5$ biological replicates) and $32 \pm 14$ s ($n = 5$ biological replicates), respectively. The relatively rapid fluorescence recovery after photobleaching suggests that, at least initially, the AMP-induced condensates exhibit liquid-like properties. To investigate this further, we have investigated AMP-induced condensate response to 1,6-hexanediol and elevated KCl concentrations. The condensates formed in the presence of P113 or Buforin-2 did not respond to 5% HD treatment, however, they did dissolve upon elevating the KCl concentration to 500 mM (Fig. 6b and Supplementary Fig. 29). Interestingly, the condensates formed in the presence of Os-C were sensitive to both 5% HD and 500 mM KCl treatment (Fig. 6b and

Supplementary Fig. 28). Based on the data, we can conclude that these condensates are liquid-like and that condensates formed in the presence of P113 or Buforin-2 are mainly held together via electrostatic interactions, whereas the Os-C condensate assembly is driven by both electrostatic and hydrophobic interactions.

Most antimicrobial peptides are generally believed to target membranes (Fig. 7 ①), while some peptides can also translocate into cells and target intracellular proteins and nucleic acids (Fig. 7 ②). Although much is known about how AMPs can perturb cell membrane integrity, the mechanisms of action of AMPs interacting with intracellular targets have remained challenging to elucidate. Here, we identify that AMPs share common biophysical properties with biopolymers that undergo phase separation suggesting that phase separation of a substantial fraction of AMPs is particularly pronounced in the presence of nucleic acids. Indeed, we show that several of these AMPs readily induce

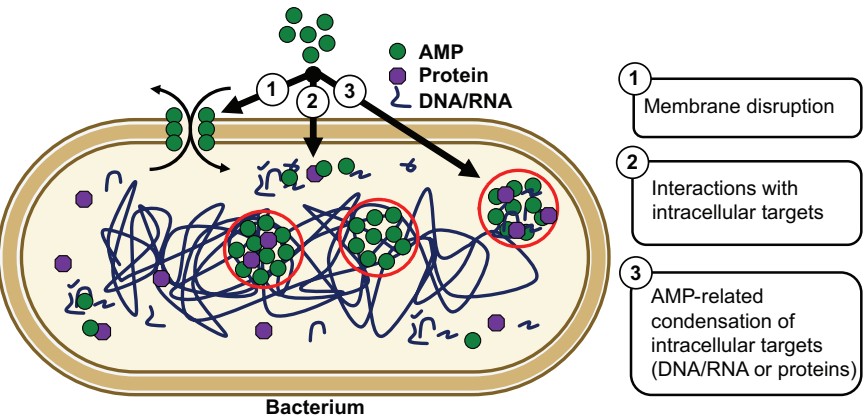

**Fig. 7 | Mechanisms of action of antimicrobial peptides.** Schematic illustration depicting known (① and ②) and proposed (③) mechanisms of actions of AMPs.

compaction and phase separation of nucleic acids in vitro. Moreover, these AMPs interfere with transcription and translation by sequestering nucleic acids. To support this model, we found AMP-rich clusters in *E. coli* that also contained bacterial nucleic acids. These observations provide evidence of an additional mode of action of DNA/RNA-targeting antimicrobial peptides: AMP-driven compaction/phase separation of nucleic acids (Fig. 7 ③). Other cellular components (e.g., nucleic acid-processing enzymes and RNA-binding proteins) may also partition into these AMP-nucleic acid condensates further exacerbating the sequestration of important biomolecules required for cell survival. These findings open exciting possibilities for the development of new anti-microbials via the engineering of AMPs that simultaneously possess high potency to induce nucleic acid compaction/phase separation and display membrane disruptive properties. Likely, combinations of membrane disruptive and RNA compacting/phase separation-prone AMPs would achieve even greater, broad-range antimicrobial activities.

## Methods
### Constructs and reagents
PBS solution was prepared by diluting 10 × RNAse-free PBS solution (Invitrogen) with nuclease-free water (Severn Biotech Ltd).

50 mM HEPES pH 7.3 solution was prepared by diluting 1 M RNAse-free HEPES solution (Fisher BioReagents™) with nuclease-free water (Severn Biotech Ltd).

16% w/v PEG (Polyethylene glycol) 20.000 Da stock solution was prepared by dissolving PEG pellets (Sigma-Aldrich) in 480 mM KCl, 3.2 mM MgCl₂, 50 mM HEPES pH 7.3 buffer solution. The stock solution was stored at room temperature.

10 mg/mL stock solutions of poly(A) RNA were prepared by dissolving poly(A) RNA powder (Sigma-Aldrich) in 50 mM HEPES pH 7.3 buffer solution. The concentration of poly(A) was determined by measuring absorbance at 260 nm using nanodrop (Implen). The stock solution was stored at −20 °C until further use. Labelled poly(A) was prepared by introducing 50 mM of SYTO-59 (Invitrogen) dye to the poly(A) stock solution.

Unlabelled and FITC-labelled TFA-free (HCl used as counter-ion) peptides were purchased from GenScript. The purity of peptides was >95%. Peptides were dissolved in 50 mM HEPES pH 7.3 buffer solution and were used in experiments without additional purification procedures. The concentration of peptides was determined by measuring UV absorbance at 280 nm or by performing Pierce rapid gold BCA protein assay (Thermo Fisher Scientific) in case of peptides lacking aromatic amino acids. Stock solutions were stored at −20 °C until further use. For experiments, unlabelled peptides were mixed with the labelled ones at a 19:1 ratio.

DNA constructs used for transcribing long RNAs, including MS2 phage RNA, 16S rRNA and 23 rRNA were transcribed as described in

Borodavka et al.[66]. Briefly, the template for transcription of RpoB RNA was produced by cloning part of the open reading frame of *E. coli* RNA Polymerase B subunit gene (*rpoB*). Templates for transcription of 16S rRNA and 23S rRNA (16SrRNA_pSMART_HCAmp and 23SrRNA_pS-MART_HCAmp) were previously produced by cloning the corresponding genes using genomic DNA extracted from *E. coli* BL21 cells (16S ribosomal RNA, GenBank: CP001665.1) and region 228583-231490 (23S ribosomal RNA, GenBank: AM946981.2) of the BL21 *E. coli* genome, and were described in detail in Borodavka et al.[66]. eGFP-coding plasmid with T7 promoter was acquired from GenScript. This was linearised by HpaI digestion to be used as a template for in vitro transcription. Transcription and fluorescent labelling of RNA in vitro transcription reactions were carried out using a T7 RNA transcription kit (HiScribe T7 High Yield; New England Biolabs) following the manufacturer's protocol. RNAs were purified using an RNeasy mini kit (QIAGEN) following the manufacturer's protocol, except for the fluorescently labelled RNAs. In those samples, the RNA-loaded column was washed four times with 80% (v/v) ethanol before elution with 30 µL of nuclease-free water. Amine-modified RNAs were produced by incorporation of amino-allyl-UTP and fluorescently labelled with Atto647N-dye, as described in Borodavka et al.[71]. All RNA samples were routinely examined on denaturing formaldehyde agarose gels to ensure their integrity. Every precaution was taken to avoid contamination with RNases, and RNA samples were kept as 10 µL aliquots at −80 °C to minimise degradation.

Template DNA harbouring gene encoding eGFP was purchased from GenScript.

### Antimicrobial peptide sequence analysis via DeePhase LLPS predictor
Antimicrobial peptide sequences deposited on the database of antimicrobial activity and structure of peptides (DBAASP)[40] were analysed using DeePhase predictor of homotypic phase separation propensity[39]. In order to estimate the phase separation propensity of protein or peptide sequence in the presence of oligonucleotides, the algorithm was retrained on data where the positive and the negative sets of sequences were those that had been seen or not seen to partition into RNA-rich condensates, respectively. These annotations were based on mass spectroscopy-based characterisation of reconstituted stress granules[72]. Importantly, as the reconstitution experiments in this study were performed on the full cell lysate, the positive set of sequences included both cytoplasmic and nuclear proteins, which prevented the algorithm from becoming too specific to either type of protein.

### Imaging of the condensates in vitro
5 mm diameter holes were punched in polydimethylsiloxane (PDMS) slabs of 3−5 mm height using a biopsy puncher. Subsequently, the

slabs were plasma bonded to a clean 24 × 60 mm No. 1.5 cover glass slides (DWK Life Sciences) to create sample imaging wells. Both imaging wells and the 18 × 18 mm glass slides (Academy) used to seal the top was treated with PEG-Silane. Briefly, the treatment solution was made by dissolving 5 mg of PEG(5000)-Silane (Sigma-Aldrich) in 20 µL of 50% acetic acid and 1 mL of ethanol. Wells and cover glass slides were treated by placing them in the solution and incubating for 1 h at 65 °C. Subsequently, they were immersed in MilliQ water and sonicated for 15 min to remove extra PEG-Silane. After washing, the wells and cover glass slides were dried under gentle airflow.

10–20 µL of samples were deposited into PEG-Silane modified sample imaging wells. The wells were covered with PEG-Silane treated cover glass slides to reduce sample evaporation. Imaging of the samples in vitro was performed using Cairn Research epifluorescence microscope (brightfield and widefield fluorescence imaging) equipped with 60 × oil immersion objective (Nikon CFI Plan Apo Lambda 60 × Oil, NA 1.4), or Leica Stellaris 5 confocal microscope (confocal fluorescence imaging) equipped with 63 × oil immersion objective (Leica HC PL APO 63×/1.40 Oil CS2, NA 1.4).

The condensates shown in Fig. 2a were formed using the following concentrations of components: 200 µM P113 and 2000 ng/µL poly(A) RNA; 200 µM Os-C and 2000 ng/µL poly(A) RNA; 400 µM Buforin-2 and 1000 ng/µL poly(A) RNA.

## Condensate fusion monitoring

Condensate fusion was monitored by following the aspect ratio, the length and width of the two fusing droplets as a function of time using a confocal microscope. The plot of aspect ratio versus the time we then fitted with the exponential decay function to find the characteristic fusion time $\tau$[73,74]. The condensates shown in Fig. 2b were formed using the following concentrations of components: 300 µM P113 and 2000 ng/µL poly(A) RNA.

## Fluorescence recovery after photobleaching (FRAP) measurements

FRAP experiments were performed using Stellaris 5 confocal microscope equipped with 63 × oil immersion objective (Leica HC PL APO 63×/1.40 Oil CS2, NA 1.4). A 488 nm argon laser at 100% power was used to bleach a disk-shaped area of 1 µm² in AMP-poly(A) condensates. The half-lives of fluorescence recovery, $\tau_{1/2}$, were obtained by analysing FRAP kymographs using the built-in software. FRAP experiments were performed on 4 different condensates for each AMP. The condensates shown in Fig. 2c were formed using the following concentrations of components: 300 µM P113 and 2000 ng/µL poly(A) RNA.

## Condensate dissolution monitoring

The P113-total yeast RNA and Oc-C-total yeast RNA condensates were made using the following concentrations of components: 375 µM P113 or OsC and 1250 ng/µL total Yeast RNA. Buforin-2-total yeast RNA condensates were made using the following concentrations of components: 500 µM Buforin-2 and 625 ng/µL total Yeast RNA. Subsequently, samples were supplemented with a 1/5 volume fraction of buffer solution, 1,6-hexanediol (25%w/v), or 2500 mM KCl in buffer solution. The final concentrations of components were 300 µM P113 or OsC, or 400 µM of Buforin-2, 1000 ng/µL total yeast RNA and 5%w/v of 1,6-hexanediol or 500 mM of KCl. Sample imaging was performed via Leica Stellaris 5 confocal microscope equipped with 63 × oil immersion objective (Leica HC PL APO 63×/1.40 Oil CS2, NA 1.4).

## Circular dichroism (CD) spectrum recording

CD spectra of 50 µM of AMPs; 125 ng/µL of poly(A) RNA, and AMP-RNA condensates in 10 mM sodium phosphate buffer solution pH 7.3 supplemented with 150 mM NaF were recorded at 20 °C using Chirascan circular dichroism spectrometer (Applied Photophysics). The spectra were recorded over a wavelength range of 240–190 nm with 1 nm

resolution and 1s time constant. For each sample, 10 spectra were recorded and an averaged spectrum was acquired. The background spectrum of the buffer solution was subtracted from all spectra.

## Construction of phase diagrams

The microfluidic devices were designed using AutoCAD software and subsequently fabricated using conventional soft-photolithography methods using SU8-on-Si wafer masters, and polydimethylsiloxane (PDMS)-on-glass devices[64,75,76]. Briefly, SU-8 3050 photoresist (A-Gas Electronic Materials Limited) was poured on a polished silicon wafer (MicroChemicals GmbH) and spun down for 45 s at 3000 RPM using a spin coater. Subsequently, SU-8 coated wafer was soft baked on a level hot plate at 95 °C for 15 min. After the soft bake step, SU-8 coated wafer was cooled down to room temperature and then the acetate sheet mask with the design of the device was placed on top of it. Mask-SU-8 coated wafer sandwich was exposed to the UV light for 40 s. Directly after the exposure, the mask was removed and the SU-8 coated wafer was post-exposure baked (PEB) on a level hot plate at 95 °C for 5 min. After PEB, the developing step took place by submerging SU-8 coated wafer into propylene glycol monomethyl ether acetate (PGMEA; Sigma-Aldrich) solution and incubating it for 8 min with periodical agitation. Finally, the wafer was rinsed with isopropyl alcohol and dried under airflow. The master wafer for fabricating microfluidic devices with a channel height of 50 µm was obtained.

The microfluidic devices were fabricated by casting PDMS (Sylgard 184 kit; Dow Corning) on a master wafer, curing it at 65 °C for 60 min, peeling it off, punching the holes for inlets and outlets, and bonding it to a 1-mm-thick glass slide (Epredia) after oxygen plasma activation in plasma oven (Diener Femto, 40% power for 30 s). Subsequently, hydrophobic treatment of the channels of the microfluidic device was performed. The channels were filled with 1% v/v trichloro(1H,1H,2H,2H-perfluorooctyl)silane (Sigma-Aldrich) in HFE-7500 fluorinated oil (3M™ Novec™ Engineered fluid) solution and incubated for 1–2 min. After the incubation, channels were washed with HFE-7500 fluorinated oil and dried under airflow.

Phase diagrams were constructed using the semi-automated microfluidic platform 'PhaseScan'[58]. Briefly, antimicrobial peptide and poly(A) stock solutions were mixed at various mass ratios (the final solutions contained 5% w/v of PEG 20,000, 150 mM KCl, 1 mM MgCl₂, 50 mM HEPES pH 7.3) and encapsulated in water-in-oil droplets, individual microenvironments, using a microfluidic device. In the case of P113 vs 23S rRNA, the final solutions contained 150 mM KCl, 1 mM MgCl₂, 50 mM HEPES pH 7.3 Fluorescence images of droplets in the observation chamber of the microfluidic device were taken using an epifluorescence microscope (Cairn Research) equipped with 10 × objective (Nikon CFI Plan Fluor 10×, NA 0.3), and analysed via automated image analysis script to detect condensates and to determine peptide and RNA concentrations in individual droplets. The data was plotted as a scatter plot with an overlayed colour-coded heat map showing the estimated phase separation probability. The phase separation probability was estimated by dividing the phase diagram into grids of bin size = 1 + 3.322 × log(total number of data points), and then dividing the total number of points labelled as phase separated by the total number of points within each grid.

## Microfluidic diffusional sizing

Microfluidic diffusional sizing of samples was performed as described previously elsewhere[64]. Briefly, first, auxiliary channels of the Fluidity One-M microfluidic chip (Fluidic Analytics) were primed with 150 mM KCl, 1 mM MgCl₂, 50 mM HEPES buffer solution pH 7.3. Subsequently, 3 × 3.5 µL of the different AMP-RNA samples were loaded into the sample channels of the microfluidic chip to measure the hydrodynamic radius of AMPs (Fig. 3) or RNA (Fig. 4). On the Fluidity One-M, the Alexa-488 detection setting and size-range setting of 1–5 nm was used for measuring the size of AMPs (Fig. 3) and the Alexa-647

detection setting and size-range setting of 5–20 nm was used to measure the size of RNA (Fig. 4).

## Measurements of AMP concentration in dilute phase at varying RNA concentrations

To perform dilute phase concentration measurements straight channel microfluidic chips were flushed with 150 mM KCl, 1 mM MgCl$_2$, 50 mM HEPES buffer solution pH 7.3 followed by equilibrating with the sample for 5 min at 100 μL/h. Flow control was performed through the application of negative pressure induced by a glass syringe (Hamilton) and connected syringe pump (neMESYS). Channel was implemented using inlet reservoirs containing buffer or sample. Dilute phase concentrations were then measured using a home-built confocal set-up[59,67]. Briefly, a 488 laser line is connected to a high magnification (60×) water-immersion objective (CFI Plan Apochromat WI 60×, NA 1.2, Nikon), where photon collection was performed using an avalanche photodiode (APD) while the microfluidic channel was mounted on a motorised XYZ stage. For further analysis, the ADP signal is binned in intervals of 1 ms to provide an intensity readout (MHz). At any time point, this is correlated to protein concentration and the dilute phase concentration can be extracted from the baseline intensity value obtained as the dense phase volume fraction can be assumed much smaller than its dilute phase equivalent. The baseline intensity and error are, hence, extracted from fitting to the histogram of collected intensities. Concentration conversion is then performed using the homogeneous, non-phase-separated reference samples.

## Determination of AMP vs RNA tie-line gradient

Calculation of tie-line gradients from dilute phase profiles was performed as described previously[67]. Briefly, the tie-line gradient is extracted from the overlapping branches of both dilute phase line scans as follows:

$$G = \frac{\Delta c_{tot}^{23S}}{\Delta c_{tot}^{P113}} \; at \; c_{dil,1}^{P113} = c_{dil,2}^{P113} \tag{1}$$

Here, $\Delta c_{tot}^{P113}$ is the difference in total P113 concentration between both line scans. To determine $\Delta c_{tot}^{23s}$ the dilute phase evolution of P113 with respect to the total 23S RNA concentration is phenomenologically fit to in the range of available data points and the expressions are set to be equal and solved for $\Delta c_{tot}^{23s}$.

## In vitro transcription and translation experiments

Cell-free eGFP synthesis was performed in NEBExpress cell-free *E. coli* protein synthesis system (New England BioLabs inc.). Briefly, the system was supplemented with the desired concentration of plasmid DNA or mRNA and AMPs, and the total volume of one reaction mixture was 50 μL. The solutions were aliquoted into 96-well plate (#3881, Corning), 50 μL per well. The plate was sealed with clear sealing tape. The plate was loaded into FLUOstar Omega (BMG Labtech) plate reader and shaken for 1 min before the fluorescence intensity measurements. The fluorescence intensity was measured every 3 min, the plate was incubated at 37 °C.

In vitro transcription reactions were carried out using a T7 RNA transcription kit (HiScribe T7 High Yield; New England Biolabs) with the following modifications. Each peptide was added to the reaction (5% v/v of the total reaction volume of 20 μL) using the eGFP DNA template containing a T7 promoter. Samples were incubated at 37 °C for 2 h, after which 1 μL of DNAseI (Turbo DNAse) was added and incubated for an additional 15 min at 37 °C. 1 μL of the reaction was diluted directly in 20 μL of formamide-containing denaturing RNA loading buffer (ThermoFisher) and incubated at 65 °C for 10 min prior to its loading on a denaturing formaldehyde agarose gel (Supplementary Fig. 16f).

To obtain reaction amplitude values the cell-free eGFP production kinetics data were fit to the simple model describing protein production kinetics in CFPS. Briefly, denote concentrations of plasmid DNA, mRNA and eGFP at time $t$ as $D(t)$, $R(t)$ and $G(t)$ respectively, we assume a simple reaction pathway with the rate laws:

$$\dot{R}(t) = k_1 D(t) \tag{2}$$

$$\dot{G}(t) = k_2 R(t) \tag{3}$$

where $k_{1,2}$ are the rate constants of mRNA and eGFP production. De-activation of the synthesis system is modelled by introducing $t$-dependence of the rates $k_1 = k_1^{(0)} e^{-t/\tau}$ and $k_2 = k_2^{(0)} e^{-t/\tau}$.

For CFPS reaction initiated with plasmid DNA:
Setting $D(t) = D = $ const. we have

$$R(t) = k_1^{(0)} D\tau (1 - e^{-t/\tau}) \tag{4}$$

which, after another direct integration, gives the eGFP concentration over time

$$G(t) = \frac{1}{2} k_1^{(0)} k_2^{(0)} D\tau^2 e^{-2t/\tau} (1 - e^{-t/\tau})^2 \tag{5}$$

At $t \to \infty$ the above goes to $G(\infty) = \frac{1}{2} k_1^{(0)} k_2^{(0)} D\tau^2$ and we can normalise the eGFP concentration according to the final value by writing

$$G(t) \equiv \frac{G(t)}{G(\infty)} = e^{-2t/\tau} (1 - e^{-t/\tau})^2 \tag{6}$$

For CFPS reaction initiated with mRNA:
Setting $R(t) = R = $ const. we have

$$G(t) = k_2^{(0)} R\tau (1 - e^{-t/\tau}) \tag{7}$$

Since the amplitude of the final protein is affected by added peptide but the initial reaction rates are not. We, therefore, suggest that the mechanism of action seems to be a reduction in the effective nucleic acid concentration that is available for the reaction. We define these relative concentrations as $R_{eff}$ and $D_{eff}$ and they are proportional to the respective starting concentrations according to the relationship:

$$R_{eff}([AMP]) = 2^{-\frac{[AMP]}{EC_{50}}} R_0 \tag{8}$$

where [AMP] is the concentration of antimicrobial peptide added to the reaction and $EC_{50}$ is the half maximal effective concentration of AMP against CFPS. The same relationship holds true for DNA. Thus, in order to extract the efficacy of each of the peptides in altering the CFPS of eGFP, we calculate the relative amplitude shift in protein expression at each AMP concentration vs. protein expression at [AMP] = 0, which allows us to extract the $EC_{50}$ for each of the peptides against mRNA and plasmid DNA reactions.

## AMP vs. Hfq competition assay

Hfq protein was purified as described elsewhere[77].

AMP vs Hfq competition assay was carried out by either incubating 100–300 μM of AMPs with 250 ng/μL of poly(A) RNA for 5 min and then introducing 0.1–300 μM of Hfq and incubating for another 5 min before imaging using a confocal fluorescence microscope (Case #1); or incubating 0.1–100 μM of Hfq with 250 ng/μL of poly(A) RNA for 5 min and then introducing 100–300 μM of AMPs and incubating for another 5 min before imaging using Leica Stellaris 5 confocal microscope equipped with 63 × oil immersion objective (Leica HC PL APO 63×/1.40 Oil CS2, NA 1.4) (Case #2).

## Cell imaging

100 mL of sterile Luria Broth (LB) media was inoculated with 20 µL of *E. coli* BL 21 DE3 cell stock and grown overnight (12–16 h) at 37 °C and 180 RPM. The next morning, 20 mL of LB was inoculated with an overnight culture so that cell optical density (O.D.) would be ~0.2 units. The cells were grown at 37 °C and 180 RPM until the cell optical density reached 0.5–0.6 units. Subsequently, cells were harvested by centrifuging for 5 min at $5000 \times g$ and resuspended in a sterile PBS so that the cell O.D. would be ~1.0 unit. Resuspended cells were mixed at a 1:1 ratio (final volume of 100 µL) with the desired concentration of Os-C, P113 or Buforin-2 peptide solution and incubated at 37 °C and 1200 RPM in benchtop shaker for 2 hours. After incubation, cells were harvested by centrifuging for 5 min at $5000 \times g$ and resuspended in sterile PBS. This step was repeated 2 times. Subsequently, cells were resuspended in 100 µL of 4% formaldehyde solution (Sigma-Aldrich) pH 6.9 supplemented with 1 × BactoView Red™ (Biotium) nucleic acid stain and incubated for 15–30 min. After incubation, cells were 3 times washed with 100 µL of PBS and finally resuspended in 20 µL of PBS.

Cells were immobilised on 24 × 60 mm No. 1.5 cover glass slides (DWK Life Sciences) as described previously[78]. Briefly, 5 µL of cells resuspended in PBS were deposited on ethanol-cleaned glass slides. 1.5% agarose (Invitrogen) in PBS pads were placed on the sample and were covered with 20 × 20 mm No. 1 cover glass slide (VWR). The open sides between two cover glass slides were sealed using VALAP solution (1:1:1 w/w Vaseline-Lanolin-Paraffin).

The samples were imaged with Leica Stellaris 5 confocal microscope equipped with 63 × oil immersion objective (Leica HC PL APO 63×/1.40 Oil CS2, NA 1.4).

## Imaging of AMP-induced condensates inside cell lysate

Extract-based active transcription/translation system was prepared by introducing 5 ng/µL DHFR-His DNA template to NEBExpress cell-free *E. coli*-based protein synthesis system (New England BioLabs inc.) and incubating the reaction mixture at 37 °C for 2 h. Subsequently, the lysate was supplemented with 300–600 µM of AMPs. The final concentrations of the lysate and AMPs were: 50% lysate and 300 µM P113; 50% lysate and 600 µM Os-C; and 10% lysate and 300 µM Buforin-2. Finally, the samples were deposited on 24 × 60 mm No. 1.5 cover glass slides (DWK Life Sciences) and imaged via Leica Stellaris 5 confocal microscope equipped with 63 × oil immersion objective (Leica HC PL APO 63×/1.40 Oil CS2, NA 1.4).

To assess the state of the condensates, initial AMP-lysate mixtures were supplemented with the same volumes of appropriate concentration lysate, 1,6-hexanediol or KCl solutions in lysate and immediately imaged via Leica Stellaris 5 confocal microscope equipped with 63 × oil immersion objective (Leica HC PL APO 63×/1.40 Oil CS2, NA 1.4).

## Statistics and reproducibility

Condensate dissolution experiments illustrated by representative images Fig. 2d and Supplementary Fig. 5 were repeated 3 times with similar results. AMP vs. Hfq competition assay (Fig. 5h and Supplementary Fig. 18 and were repeated 3 times with similar results. Experiments illustrating AMP-induced formation of foci-like condensates in *E. coli* (Fig. 6a and Supplementary Figs. 20–27) were repeated 2 times with similar results. Experiments illustrating AMP-induced formation of foci-like condensates in *E. coli* lysate (Fig. 6b and Supplementary Figs. 28–29) were repeated 3 times with similar results.

## Reporting summary

Further information on research design is available in the Nature Portfolio Reporting Summary linked to this article.

## Data availability

All data generated or analysed during this study are included in this published article (and its supplementary information files). The data acquired from the Database of Antimicrobial Activity and Structure of Peptides can be accessed via the following link: https://dbaasp.org/search. Structural statistics of AMPs in the APD3 antimicrobial peptide database can be accessed via the following link: https://aps.unmc.edu/statistic/structure. Source data are provided with this paper.

## Code availability

The custom codes used in a current study are available at GitHub: https://github.com/rqi14/PhaseScan and https://github.com/kadiliissaar/deephase.

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

## Acknowledgements
The research leading to these results has received funding from the European Union's Horizon 2020 research and innovation programme under the Marie Skłodowska-Curie grant MicroREvolution (agreement no. 101023060; T.S.) and the ERC grant DiProPhys (agreement ID 10100161, T.P.J.K.); the Royall Scholarship (N.A.E.); Global Research Technologies Novo Nordisk A/S (H.A., T.P.J.K.); the European Research Council under the European Union's Seventh Framework Programme (FP7/2007-2013) through the ERC grants PhysProt (agreement no. 337969; T.P.J.K.); the Frances and Augustus Newman Foundation (T.S.); the Schmidt Science Fellowship programme in partnership with the Rhodes Trust (K.L.S.); St. John's College Junior Research Fellowship (K.L.S.), the Harding Distinguished Postgraduate Scholar Programme (T.J.W.); Boehringer Ingelheim Fonds (K.K.G.); and Wellcome Trust (213437/Z/18/Z, A.B.).

## Author contributions
T.S., N.A.E., H.A., A.B and T.P.J.K. conceived the study. T.S., N.A.E., H.A., D.Q., M.L.L.Y.J, K.L.S., T.J.W., K.K.G, G.K. and A.B. performed the investigation. A.B. and T.P.J.K. provided resources. T.P.J.K. acquired funding. T.S., N.A.E., H.A., A.B. and T.P.J.K. wrote the original draft, and all authors reviewed and edited the paper.

## Competing interests
The Authors declare the following competing interests: T.P.J.K. is a co-founder and H.A., K.L.S., D.Q., G.K. and T.P.J.K. are employees or consultants for Transition Bio; T.S., N.A.E., M.L.L.Y.J., T.J.W., K.K.G. and A.B. declare no competing interests.
