## [Peer Review File · Nature Communications]

REVIEWER COMMENTS

Reviewer #1 (Remarks to the Author):

In this work, Knowles and colleagues propose an interesting angle of how anti-microbial peptides may work by initiating condensation of DNA and RNA in bacterial cells. The authors first perform bioinformatic analysis based on their previous work. Subsequently, they chose three AMPs and showed that all three peptides form condensates with polyA RNA in vitro. Using microfluidic-based analysis, the authors study their phase diagram. They also perform functional test of AMPs using cell free system of transcription and translation processes. Finally, they show that these AMPs form puncta like structures in bacterial cells. I have a few comments below.

Overall, the authors have a very interesting hypothesis, but in my opinion, they fall short in demonstrating that AMPs actually work this way. The work seems to be preliminary, more experimental evidence is needed. The peptides are very short; therefore, a major question is how are they competing against the native RNA/DNA binding proteins? This particular point (the peptide length) also makes the bioinformatic analysis questionable, specially, when they compare AMPs with human proteins. Furthermore, generalization that AMPs function this way (by modulating RNA/DNA condensation) is problematic without further testing since it is unclear if all cationic peptides can work this way. My take of this manuscript is that it is suitable for a more focused journals rather than the broad readership of Nat Commun at this current stage, however, it may become suitable for Nat Commun after rigorous studies are included. There is no surprise that cation multivalent peptides will phase separate with polyA RNA but connecting such a property with anti-microbial needs further considerations. The proposed model is novel, nonetheless.

Reviewer #2 (Remarks to the Author):

In the manuscript entitled "Modulating Nucleic Acid Phase Transitions as a Mechanism of Action for Cell-Penetrating Antimicrobial Peptides" by Sneideris, T. et al., the authors report about the capability of antimicrobial peptides in inducing the phase separation of nucleic acids (specifically, RNAs). The liquid-like nature of droplets was well demonstrated. Then, the authors proposed the observed phenomenon as a possible action mechanism for cell-penetrating peptides.

The manuscript is really well written and clear. The authors used several experimental techniques and machine learning algorithms to predict the phase separation behavior of peptides with and without nucleic acids (RNAs). Finally, microscopy experiments on bacteria were also carried out to support their

findings. All the experiments seem well conducted and with high scientific rigor. However, the conclusion of the study is not fully supported by experimental evidences.

The ability of AMPs to affect LLPS is not a new topic. Indeed, several papers are present in the literature (for example (doi): 10.1016/j.molcel.2017.02.013; 10.1039/D0CC04877A) However, in the cited papers, the effects of AMPs on LLPS of proteins are described and are more focused on the world of eukaryotes. Thus, AMPs-induced phase separation of RNAs as a possible antibacterial action mechanism is new, interesting and of high relevance/significance.

Before to support its publication on Nature Communications, several important points should be addressed:

- 1) The authors mentioned Cell-Penetrating peptides in the title. However, no description of them is reported in the main text. The authors should report on them highlighting the differences with common AMPs (e.g. the presence of arginines instead of lysines; some of the are able to translocate across the membrane without affecting its integrity,).
- 2) It is known that almost all linear cationic peptides assume helical conformation upon binding membranes. What is the conformation of the three used AMPs within the droplets?
- 3) In Figure 1 on the y-axis it is reported "Nucleic acid". In the methods section, I have seen that the analysis was performed using RNAs. So, I would suggest to clearly indicate that it refers to RNAs.
- 4) The phase diagrams reported in Figure 3 are really informative. As stated by the authors, P113 peptide is the most effective in inducing LLPS formation, since the concentrations of the components are lower respect to the other systems. I strongly believe that this could be due to the different affinity of the peptides for polyA, i.e. the complex formation before the LLPS formation (since the three AMPs have the same charge, this will indicate that the primary sequence of peptides plays a key role). Thus, binding experiments for the determination of dissociation constants (outside the LLPS regime) should be performed. In addition, in Figure 4 panel e, the authors reported a schematic of pairwise interaction whose proof is missing. Thus, with binding experiments the authors could prove the existence of pairwise interaction and, at the same time, explaining the different behaviors (concentrations required for LLPS formation) reported in the phase diagrams.
- 5) In bacteria, the observed foci-like structures are really LLPS? This is the most important point of the study. To claim that AMPs under study can really act through the new proposed mechanism, the authors should demonstrate that these are really LLPS. In my opinion, the structures observed can be also aggregates. With this proof, I would be really happy to support the publication of this manuscript. Just as suggestion, in one of the cited references (23 of the manuscript), the authors proved LLPS formation using 1,6-hexanediol.
- 6) Page 5, line 139 the authors stated "Notably, the average size of droplets increased with higher peptide/RNA concentration", where is it shown?

7) Figure 2 panel B, please report at which AMP:polyA ratio refers the pictures.

8) I found a bit difficult to follow Figure 5 (a-f). Maybe, instead of colors, the authors could use different symbols.

9) I really appreciate the experiments reported in Figure 5. However, it is not clear to me why Bufenin which is capable to induce phase separation of RNAs, cannot affect the translation process, even at high concentrations (as seen in other cases). Have the authors an explanation for this?

10) Please, check again the manuscript, some errors are present in the text (e.g. page 7, line 196, structured is repeated).

Response letter

Reviewer #1 (Remarks to the Author):

Reviewer 1: In this work, Knowles and colleagues propose an interesting angle of how anti-microbial peptides may work by initiating condensation of DNA and RNA in bacterial cells. The authors first perform bioinformatic analysis based on their previous work. Subsequently, they chose three AMPs and showed that all three peptides form condensates with polyA RNA in vitro. Using microfluidic-based analysis, the authors study their phase diagram. They also perform functional test of AMPs using cell free system of transcription and translation processes. Finally, they show that these AMPs form puncta like structures in bacterial cells. I have a few comments below.

Response: We thank the reviewer for their positive assessment and for the thoughtful comments provided.

Reviewer #1: Overall, the authors have a very interesting hypothesis, but in my opinion, they fall short in demonstrating that AMPs actually work this way. The work seems to be preliminary, more experimental evidence is needed. The peptides are very short; therefore, a major question is how are they competing against the native RNA/DNA binding proteins?

Response: We thank the reviewer for raising this good point. Inspired by this comment, we have performed a set of additional experiments to investigate how AMPs compete against a ubiquitous RNA-binding protein (RBP) Hfq highly abundant in bacterial cells. Interestingly, our new experimental results indicate that once AMP-RNA condensates are formed, they are very stable and high RBP concentrations are required to disrupt the condensates by outcompeting AMPs. In a similar manner, at varying AMP:RBP molar ratios, AMPs can induce phase separation of RNA. We added the results from these new experiments to our revised paper (Figure 5h and Figures S18-19, lines 318-331) and thank the referee for prompting us to carry out these additional useful experiments.

Reviewer #1: This particular point (the peptide length) also makes the bioinformatic analysis questionable, specially, when they compare AMPs with human proteins.

Response: As the referee points out, the average peptide length is shorter than the average length of proteins in the human proteome. To eliminate any bias due to length differences between AMPs and the human proteome on the phase separation propensity profiles, we additionally screened two sets of 10,000 unique random AA sequences of variable lengths. The results show that the chain length alone is unlikely to account for such distinct phase separation propensity profiles, further suggesting that the observed differences may be attributed to the unique AA composition of AMPs (Figure S1, lines 125-130).

Reviewer #1: Furthermore, generalization that AMPs function this way (by modulating RNA/DNA condensation) is problematic without further testing since it is unclear if all cationic peptides can work this way.

Response: This is a good point. In our paper, we look at specific examples of such systems, but more recently further independent confirmation of this general phenomenon has emerged

[\[https://doi.org/10.1101/2023.03.09.531820\]](https://doi.org/10.1101/2023.03.09.531820) suggesting that this is likely to be a more general mechanism.

Reviewer #1: My take of this manuscript is that it is suitable for a more focused journals rather than the broad readership of Nat Commun at this current stage, however, it may become suitable for Nat Commun after rigorous studies are included. There is no surprise that cation multivalent peptides will phase separate with polyA RNA but connecting such a property with anti-microbial needs further considerations. The proposed model is novel, nonetheless.

Response: We thank the referee for highlighting the novelty of the model and mechanism. We hope that the additional experiments that we have carried out further help to clarify the points that the referee raised and thank them again for taking the time to look at our paper and suggest experiments.

Reviewer #2 (Remarks to the Author):

Reviewer #2: In the manuscript entitled “Modulating Nucleic Acid Phase Transitions as a Mechanism of Action for Cell-Penetrating Antimicrobial Peptides” by Sneideris, T. et al., the authors report about the capability of antimicrobial peptides in inducing the phase separation of nucleic acids (specifically, RNAs). The liquid-like nature of droplets was well demonstrated. Then, the authors proposed the observed phenomenon as a possible action mechanism for cell-penetrating peptides.

The manuscript is really well written and clear. The authors used several experimental techniques and machine learning algorithms to predict the phase separation behavior of peptides with and without nucleic acids (RNAs). Finally, microscopy experiments on bacteria were also carried out to support their findings. All the experiments seem well conducted and with high scientific rigor.

Response: We thank the reviewer for their time, and for highlighting the high scientific rigor of our experiments, as well as their recognition of the clarity of the manuscript.

Reviewer #2: However, the conclusion of the study is not fully supported by experimental evidences. The ability of AMPs to affect LLPS is not a new topic. Indeed, several papers are present in the literature (for example (doi): 10.1016/j.molcel.2017.02.013; 10.1039/D0CC04877A) However, in the cited papers, the effects of AMPs on LLPS of proteins are described and are more focused on the world of eukaryotes. Thus, AMPs-induced phase separation of RNAs as a possible antibacterial action mechanism is new, interesting and of high relevance/significance.

Response: We thank the reviewer for highlighting the novel nature of the mechanism and the high interest and significance of the findings. We agree that the literature on the ability of AMPs to modulate LLPS behaviour of eukaryotic proteins is highly relevant, and we have expanded the discussion and cited these papers in our revised paper (Lines 45-48). What is exciting about the present work is that it extends these ideas to bacterial condensates including nucleic acids.

Reviewer #2: Before to support its publication on Nature Communications, several important points should be addressed:

The authors mentioned Cell-Penetrating peptides in the title. However, no description of them is reported in the main text. The authors should report on them highlighting the differences with common AMPs (e.g. the presence of arginines instead of lysines; some of the are able to translocate across the membrane without affecting its integrity, ...).

Response: We thank the reviewer for the following comment. We have included additional text highlighting the few known characteristics of cell-penetrating antimicrobial peptides to address this comment (Lines 35-40).

Reviewer #2: It is known that almost all linear cationic peptides assume helical conformation upon binding membranes. What is the conformation of the three used AMPs within the droplets?

Response: We thank the reviewer for raising this question. We have recorded CD spectra of AMPs in a condensed state (Figure 2e and Figure S6, lines 168-176). The results suggest that AMPs do not acquire any structure and remain mainly disordered. The findings are in line with the hypothesis that, at least in freshly formed condensates, disordered proteins or low-complexity domains within the phase-separating proteins remain disordered, which allows them to dynamically interact with each other inside the condensates.

Reviewer #2: In Figure 1 on the y-axis it is reported "Nucleic acid". In the methods section, I have seen that the analysis was performed using RNAs. So, I would suggest to clearly indicate that it refers to RNAs.

Response: We have modified the y-axis label as suggested by the reviewer.

Reviewer #2: The phase diagrams reported in Figure 3 are really informative. As stated by the authors, P113 peptide is the most effective in inducing LLPS formation, since the concentrations of the components are lower respect to the other systems. I strongly believe that this could be due to the different affinity of the peptides for polyA, i.e. the complex formation before the LLPS formation (since the three AMPs have the same charge, this will indicate that the primary sequence of peptides plays a key role). Thus, binding experiments for the determination of dissociation constants (outside the LLPS regime) should be performed. In addition, in Figure 4 panel e, the authors reported a schematic of pairwise interaction whose proof is missing. Thus, with binding experiments the authors could prove the existence of pairwise interaction and, at the same time, explaining the different behaviors (concentrations required for LLPS formation) reported in the phase diagrams.

Response: We thank the reviewer for this excellent suggestion. We have performed AMP-poly(A) binding experiments outside the LLPS regime and found that neither of the peptides displays a high poly(A)-binding affinity (Figure 3f, lines 201-211). However, out of the three peptides investigated, P113 has the strongest relative affinity whereas Buforin-2-the weakest. Thus, the AMPs with slightly higher affinity for a particular RNA have a higher phase separation propensity. The overall low affinity for RNA favours the formation of condensates as the assembly of biomolecular condensates is enabled by weak but collective interactions.

Reviewer #2: In bacteria, the observed foci-like structures are really LLPS? This is the most important point of the study. To claim that AMPs under study can really act through the new proposed mechanism, the authors should demonstrate that these are really LLPS. In my opinion, the structures observed can be also aggregates. With this proof, I would be really happy to support the publication of this manuscript. Just as suggestion, in one of the cited references (23 of the manuscript), the authors proved LLPS formation using 1,6-hexanediol.

Response: We thank the referee for raising this good point. To better characterise the nature of sub-micron-sized foci-like structures that we observed inside *E. coli*, we have interrogated the AMP-induced condensates formed in bacterial cell lysates (Figures 7 and S28-29, lines 366-386). In this system, we can directly address the questions posed by the referee, including the addition of phase separation modulators like 1,6-hexanediol (HD) or change the ionic strength (by adding KCl), and perform FRAP experiments to check the fluidity of the condensates (Lines 373-386). The FRAP experiments showed rapid recovery post-bleaching (Figure S30). Furthermore, the condensates rapidly dissolved upon elevating the KCl concentration to 500 mM. The condensates formed in the presence of Os-C were sensitive to both 5% HD and 500 mM KCl treatment, but the other systems were only sensitive to the addition to KCl but not HD, pointing towards a key role of electrostatic interactions in stabilising the condensates.

Reviewer #2: Page 5, line 139 the authors stated “Notably, the average size of droplets increased with higher peptide/RNA concentration”, where is it shown?

Response: This type of behaviour is commonly observed when the dense phase volume fraction increases when moving deeper into the two-phase region in the phase diagram of a liquid-liquid phase separating system. We have some preliminary observations which are in agreement with this idea for our system, but since this is not surprising and not essential for the message of the paper, we have removed the sentence in question to avoid any confusion.

Reviewer #2: Figure 2 panel B, please report at which AMP:polyA ratio refers the pictures.

Response: We have provided the following information in the caption of Figure 2 as requested.

Reviewer #2: I found a bit difficult to follow Figure 5 (a-f). Maybe, instead of colors, the authors could use different symbols.

Response: We thank the reviewer for the suggestion. To make Figure 5 more clear, in addition to using different colours, we used different symbols to separate distinct data sets.

Reviewer #2: I really appreciate the experiments reported in Figure 5. However, it is not clear to me why Bufenin which is capable to induce phase separation of RNAs, cannot affect the translation process, even at high concentrations (as seen in other cases). Have the authors an explanation for this?

Response: This is an excellent comment which has inspired us to conduct additional experiments. Bufenin-2 does affect transcription and translation, however, to a much lesser extent compared to the other two peptides investigated. To better show this, we have included additional data where the addition of 600 μ M of Bufenin-2 into the cell-free protein synthesis reaction results in a more pronounced inhibitory effect (Figure 5).

Reviewer #2: Please, check again the manuscript, some errors are present in the text (e.g. page 7, line 196, structured is repeated).

Response: Thank you for pointing out these issues. We have carefully and thoroughly checked the manuscript for errors, these changes are now incorporated in the revised manuscript.

REVIEWERS' COMMENTS

Reviewer #3 (Remarks to the Author):

In the revised version of the manuscript entitled “Modulating Nucleic Acid Phase Transitions as a Mechanism of Action for Cell-Penetrating Antimicrobial Peptides” by Sneideris, T. et al., the authors have addressed the concerns raised by this reviewer. The authors have performed new experiments supporting their findings. In particular, I appreciate experiments on cell extracts (page 13). The obtained results strongly support the hypothesis that the foci-like structures seen in bacteria are effectively biocondensates. This is the key point of the study. Overall, I found the manuscript substantially improved respect to the first version.

Minor point:

I appreciate also the tentative measurements of peptides' affinity for RNA (tentative because no dissociation constants are reported). However, some qualitative conclusions can be reached. It seems to me that the affinity for RNA follows the order P113 > Os-s > Buforin (comparing the Rh values at the highest RNA concentration). This trend would match perfectly with the phase separation behaviour of peptides. Maybe, some experimental points at higher RNA concentration could better highlight the differences. Finally, I suggest to remove the sentence at page 8 line 207 stating that the affinity is low. It is not possible to declare this without having the values of the constant.

I want to highlight that this is only a minor point of the work. With all the data provided by the authors, their conclusions are well supported. Thus, I can only recommend its publication on Nature Communications.

Response to the Reviewers

Reviewer #2: I appreciate also the tentative measurements of peptides' affinity for RNA (tentative because no dissociation constants are reported). However, some qualitative conclusions can be reached. It seems to me that the affinity for RNA follows the order P113 > Os-s > Buforin (comparing the R_h values at the highest RNA concentration). This trend would match perfectly with the phase separation behaviour of peptides. Maybe, some experimental points at higher RNA concentration could better highlight the differences. Finally, I suggest to remove the sentence at page 8 line 207 stating that the affinity is low. It is not possible to declare this without having the values of the constant.

Response: We thank the reviewer for a great suggestion. We revised the paragraph according to the suggestion.